# Enzymatic synthesis of key RNA therapeutic building blocks using simple phosphate donors

Qinglong Meng[1], Caecilie Benckendorff [2], Charlotte Morrill[1], Ying Zhuo[1], Annette Egerström [1], Aisling Ní Cheallaigh[2], Sasha R. Derrington[1], Richard Obexer [1], Mary Ortmayer[1], Colin W. Levy[1], James D. Finnigan [3], Simon J. Charnock[3], Nicholas J. Turner [1], Gavin J. Miller [2] & Sarah L. Lovelock [1] ✉

The rapid emergence of RNA therapeutics has highlighted the need for more efficient, scalable and sustainable methods for their manufacture. Biocatalytic approaches hold particular promise, but rely on a secure, sustainable and low-cost supply of nucleoside triphosphate (NTP) building blocks, including those containing chemical modifications. Here we report the development of a biocatalytic approach and engineered enzymes to convert widely available nucleosides into NTPs featuring pharmaceutically relevant modifications using inexpensive phosphate donors. Importantly our strategy obviates the need for ATP as a phosphate donor that complicates NTP isolation using existing methods. To showcase the utility of our approach, we employ an engineered acid phosphatase, polyphosphate kinase and acetate kinase to produce 2′-O-methoxyethyl-ATP (2′-MOE-ATP) and 2′-fluoro-ATP, key building blocks of commercial therapeutics. Finally, we show that crude NTPs from our process can be used directly in enzymatic oligonucleotide synthesis, obviating the need for costly NTP isolation or purification steps.

Recent years have seen rapid expansion in the field of RNA therapeutics[1,2]. For example, mRNA vaccines were developed in response to the COVID-19 pandemic, and mRNA therapeutics are now emerging for the treatment of cancers and genetic disorders[3–7]. An increasing number of antisense oligonucleotides (ASOs) and small-interfering RNAs (siRNAs) have also been approved and many more are in advanced stages of clinical trials[8,9]. To address the increasing demand for high volumes of RNA products, more efficient and scalable synthetic methods are urgently needed[10,11]. In this regard, several complementary biocatalytic strategies have been developed using terminal deoxyribonucleotidyl transferases (TdTs)[12–15], RNA-[16,17], DNA-[18], or Poly(U)- polymerases[19]. A common feature of these approaches is the requirement for nucleoside triphosphates (NTPs) as monomeric building blocks. However, NTPs are notoriously challenging to chemically synthesize due to poor solubility in organic solvent and the presence of multiple reactive groups on the ribose sugar and nucleobases. The challenges of efficient NTP synthesis are amplified by the common requirement for chemical modifications to the ribose sugar, nucleobase and/or phosphate backbone, which confer beneficial pharmacokinetic and pharmacodynamic properties[2,9,20]. Traditional P(III)-based approaches such as the Yoshikawa and Ludwig-Eckstein methods work well for some NTPs, but are generally hampered by low yields, the need for moisture- and air-sensitive reagents, the requirement for protecting groups and complex purifications[21,22]. Alternative approaches involving the use of derivatized P(III) phosphoramidites[23–25], P(V) organic pyrophosphates[26] or triphosphate transfer reagents[27] offer some advantages and facilitate isolation/purification of NTP products.

[1]Manchester Institute of Biotechnology and Department of Chemistry, University of Manchester, Manchester, UK. [2]Centre for Glycoscience and School of Chemical and Physical Sciences, Keele University, Keele, Staffordshire, UK. [3]Prozomix Ltd, Haltwhistle, UK. ✉e-mail: sarah.lovelock@manchester.ac.uk

However, these methods rely on specialized reagents that are produced through multi-step chemical synthesis. As a result, production of modified NTPs in a scalable, economical and sustainable fashion remains a major challenge.

Beyond enzymatic synthesis of RNA therapeutics, biocatalysis could offer a more efficient route to the requisite NTP building blocks. In nature, nucleosides can be sequentially phosphorylated through the action of kinases that transfer a γ-phosphate from ATP to nucleoside, nucleoside monophosphate (NMP), or nucleoside diphosphate (NDP) substrates. This approach has formed the basis of biocatalytic strategies to produce NTPs (Fig. 1A)[28–34]; however these kinases have not been shown to operate on many substrates featuring common 2′-ribose modifications or 3′- protecting groups that are required for enzymatic synthesis of oligonucleotide therapeutics. Moreover, the use of ATP as a phosphate donor raises several challenges. ATP is expensive, necessitating the use of ancillary enzymes for cofactor recycling that must operate in conjunction with the three kinases required for NTP synthesis. Furthermore, when producing NTPs that are close structural analogues of ATP (e.g., 2′-flouro ATP), the presence of ATP and ADP by-products complicates downstream purification. Given that ATP is commonly an excellent substrate for downstream enzymes used during oligonucleotide synthesis[16–19], even low levels of ATP impurities can result in heterogeneous product mixtures. An elegant solution to avoid the requirement for ATP involves the development of engineered kinases that can use catalytic concentrations of target NTP products as a phosphate donor in conjunction with an appropriate recycling system[29]. This approach was recently used to produce 2′-F-(Sp)-thioATP for production of a cyclic dinucleotide analogue[29], however, it unknown whether cofactor specificity can be more extensively reprogrammed towards alternative bases and/or ribose modifications.

To address existing limitations of NTP synthesis, here we establish a versatile ATP-free biocatalytic approach (Fig. 1B) that is suitable for preparative scale synthesis and enables production of a wide variety of therapeutically relevant NTP building blocks.

## Results and discussion

Our proposed biocatalytic strategy involves three enzymatic transformations. The first step makes use of a non-specific acid phosphatase to install a 5′-monophosphate onto commercially available nucleosides[35–38]. The resulting NMPs can be converted under thermodynamic control to mixtures of NDPs and NTPs using a polyphosphate kinase (PPK). While PPKs have previously been employed in ATP recycling systems and in the generation of base modified NTPs[39–42], they have not yet been shown to tolerate 2′- and 3′-ribose modifications. Finally, an acetate kinase (AcK) can be used to phosphorylate any remaining NDP to improve yields of the target NTPs[29,31]. In principle, this strategy should allow us to achieve our goal of realizing ATP-free NTP synthesis to underpin future biocatalytic production of RNA therapeutics. However, successful development of such a platform requires the discovery and/or engineering of biocatalysts that can operate on nucleosides featuring diverse chemical modifications that are commonly found in therapeutic sequences. Ideally, these enzymes would also be able to operate in a telescoped process, where consecutive biotransformations are performed without isolation or purification of intermediates.

### An engineered acid phosphatase (PhoC)

We began by selecting an engineered non-specific acid phosphatase from *Morganella morganii* containing G92D and I171T mutations (PhoC_1) that had previously been shown to reduce competing enzyme catalyzed hydrolysis of 5′-phosphorylated nucleoside products[35]. This variant has been used to prepare 5′-inosine monophosphate on a 100 g scale, making it an attractive candidate for further development as an industrial biocatalyst for modified NTP production[35]. However, despite its potential there is little known about the ability of this enzyme to operate on chemically modified nucleosides as required for our target applications. Indeed, initial activity assays performed on a small panel of commercial nucleosides reveal that enzyme activity is severely compromised on substrates containing modifications that are commonly found in therapeutic sequences (see below). To overcome these limitations, we elected to broaden the substrate tolerance of PhoC_1 through enzyme engineering.

We selected 2′-O-methoxyethyl adenosine (2′-MOE-A, **1a**) as a target substrate for evolutionary optimization using pyrophosphate as a phosphate donor (Fig. 2A). 2′-MOE modified nucleotides are found in approved oligonucleotide therapeutics, including mipomersen and inotersen, yet the corresponding 2′-MOE-NTP building blocks are not

**Fig. 1 | Biocatalytic approaches to NTPs. A** An ATP-dependent enzyme cascade employing a nucleoside kinase (NK), nucleoside monophosphate kinase (NMPK) and nucleoside diphosphate kinase (NDPK). **B** An ATP independent enzyme cascade employing an acid phosphatase (PhoC), polyphosphate kinase (PPK) and acetate kinase (AcK). NMP nucleoside monophosphate, NDP nucleoside diphosphate, NTP nucleoside triphosphate.

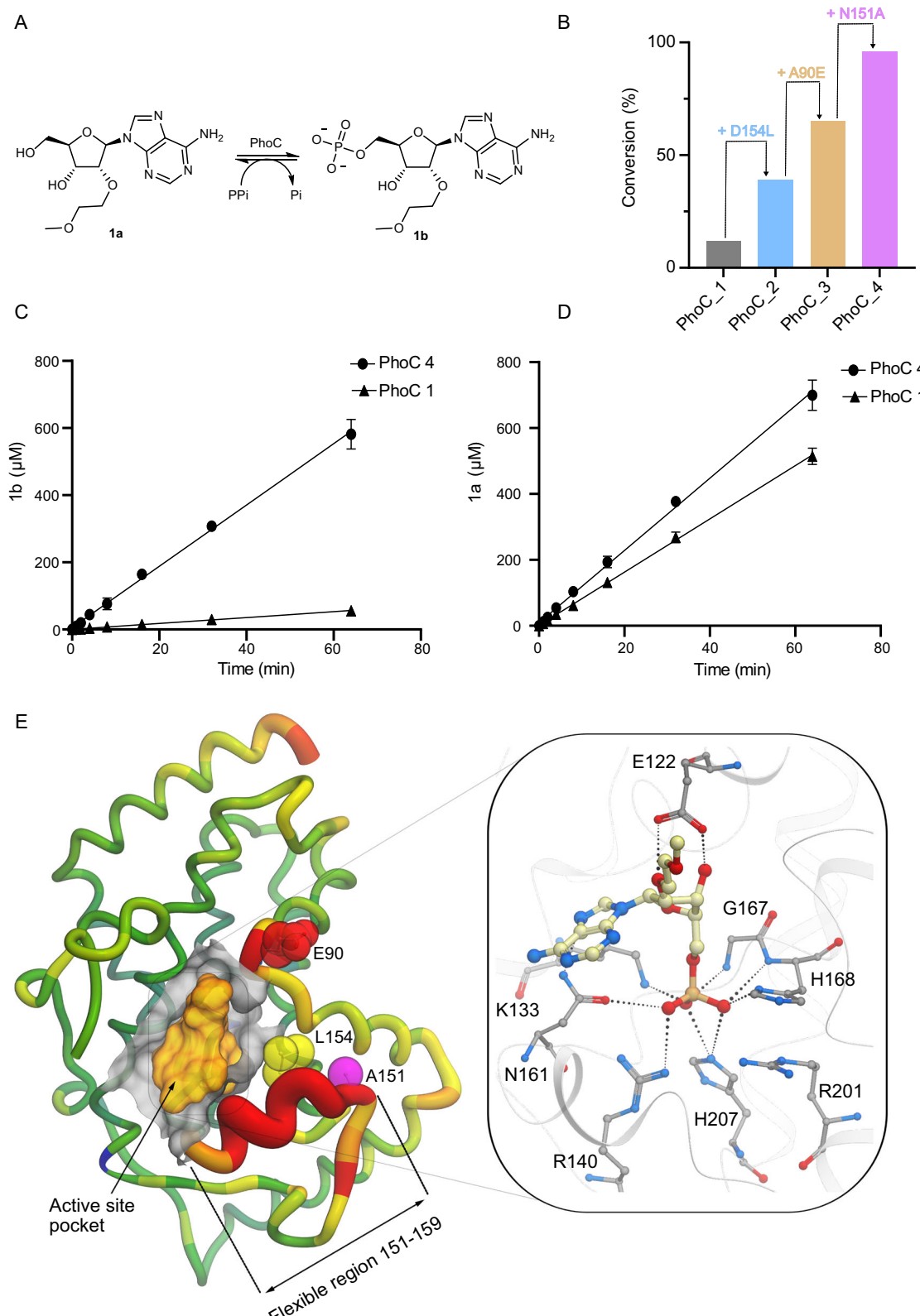

widely available from commercial suppliers[2,9,10]. The evolutionary strategy consisted of two rounds of saturation mutagenesis targeting residues in the active site, second coordination sphere and flexible loop regions. In each round 20 positions were individually randomized using NNK degenerate codons (theoretical library size of 640 genetic variants per round). Individual variants were arrayed as clarified cell lysates in 96-well microtitre plates and evaluated using a high-throughput UPLC assay monitoring the conversion of **1a** to **1b**. The most promising (ca. 1%) clones were then evaluated as purified proteins for improved activity (Supplementary Fig. 1). Beneficial mutations identified during each round were combined by overlap extension PCR. The most active variant (PhoC_4) to emerge following evaluation of more than 3500 clones contains three mutations (A90E, N151A and D154L) and displays a substantial eightfold improvement in

**Fig. 2 | Characterization of PhoC_1 and variants. A** Chemical scheme of the phosphorylation reaction of **1a** catalyzed by PhoC; **B** Bar chart showing the percentage conversion achieved by selected variants along the evolutionary trajectory. Biotransformations were performed using **1a** (1 mM), pyrophosphate (200 mM) and enzyme (20 μM) in 20 mM sodium acetate buffer pH 4 at 30 °C. **C** Initial rate of **1a** phosphorylation by PhoC_1 and PhoC_4 using pyrophosphate. Biotransformations were performed using **1a** (1 mM), pyrophosphate (200 mM) and enzyme (5 μM) in 20 mM sodium acetate buffer pH 4 at 30 °C. The concentration of product **1d** (mean ± s.d.) was measured from reactions performed in triplicate. **D** Initial rate of **1b** hydrolysis by PhoC_1 and PhoC_4 in the absence of a phosphate donor. Biotransformations were performed using **1b** (1 mM) and enzyme (0.1 μM) in 20 mM sodium acetate buffer pH 4 at 30 °C. The concentration of product **1a** (mean ± s.d.) was measured from reactions performed in triplicate. **E** PhoC_4* is shown in protein worm representation, coloured and scaled by B-factor highlighting the flexible nature of the active site pocket in the vicinity of the inserted mutations. Mutations installed during the course of evolution are highlighted with CPK spheres N151A (purple), D154L (yellow) & A90E (red). The active site pocket is highlighted with a grey coloured surface representation of the protein and an orange space filling envelope of a docked 2′-MOE-AMP product. The right-hand panel shows a close up of the docked 2′-MOE-AMP shown in ball and stick representation and coloured by atom type. Key protein interactions are highlighted (ball and stick) in addition to hydrogen bonds (dotted lines) and the protein backbone (ribbon representation). Source data for (**B–D**) are provided as a source data file.

the conversion of **1a** to **1b** under the assay conditions used during evolution (96% and 12% conversion with PhoC_4 and PhoC_1, respectively) (Fig. 2B). This improvement arises from a tenfold increase in the rate of **1a** phosphorylation, alongside a more modest 1.4-fold increase in the rate of competing 2′-MOE-AMP (**1b**) hydrolysis, showing how the mutations installed during evolution have led to a more favourable partitioning of PhoC activities (Fig. 2C, D). Kinetic analysis shows PhoC_4 displays a ninefold improvement in catalytic efficiency ($k_{cat}/K_M$) compared with the parent enzyme PhoC_1. These improvements are attributed to a 3.5-fold reduction in $K_M$ and a threefold improvement in $k_{cat}$ (PhoC_1 $k_{cat} = 0.3$ s$^{-1}$; PhoC_4 $k_{cat} = 0.8$ s$^{-1}$, Supplementary Fig. 2).

To understand the role of mutations installed during evolution, PhoC_1 and PhoC_4 were both crystallized and diffracted to 2.2 and 2.0 Å, respectively. Crystals formed in space group P3$_2$12 with three molecules present in the asymmetric unit. Expansion of the crystal symmetry, supported by PISA analysis[43,44], indicates a stable hexameric quaternary structure (trimer of dimers) similar to that observed in the acid phosphatase from *Escherichia blattae* (pdb codes 1D2T & 1EOI)[45]. Individual subunits from PhoC_1 and PhoC_4 can be superimposed with an RMSD of 0.15 Å (Supplementary Fig. 3). A comparative analysis of PhoC_4 with 1D2T and 1EOI suggests that the three mutations installed during PhoC_4 evolution (A90E, N151A and D154L) all reside in dynamic regions of the structure. N151A and D154L, reside on a poorly resolved loop region (residues 151–159) similar to that observed in 1D2T (residues 133–143). In 1EOI, which contains molybdate as a transition state analogue, this loop undergoes a structural rearrangement into a helical architecture to create a more occluded binding pocket. Based upon sequence and structural alignment, A90E in PhoC_4 is equivalent to residue S72 in 1EOI, which undergoes a 6 Å reorientation upon molybdate binding. In the absence of a substrate or product-bound PhoC_4 complex, docking studies were performed on a PhoC_4 model (PhoC_4*) generated using molecular dynamics to produce a conformation similar to that observed in 1EOI[45]. The observed docking pose of 2′-MOE-AMP and key interactions are presented in Fig. 2E. In this model, Arg201 and His168 adopt alternative conformers to those observed in the apo-structures to facilitate phosphate binding through polar interactions. Replacement of His168 by alanine in PhoC_4 led to a 48-fold reduction in activity, while mutation of Arg201 to alanine completely abolishes activity. Although further studies are needed to elucidate the specific role of the polar to hydrophobic N151A and D154L mutations, this analysis suggests that the flexible loop region 151–159 plays an important role in substrate recognition and/or the partitioning of phosphoryl transfer vs hydrolysis activity, marking this region as an attractive target for more extensive engineering to develop more proficient biocatalysts.

With an improved PhoC variant in hand, we next evaluated its activity towards a diverse panel of canonical and modified nucleosides (**1a–32a**, Fig. 3). Compared with the PhoC_1 starting variant, PhoC_4 displays improved efficiency towards all substrates tested. In addition to 2′-MOE-A (**1a**), PhoC_4 displays high levels of activity towards other 2′-MOE nucleosides (**20–22a**) as well as substrates with other

therapeutically important 2′-substituents including 2′-fluoro- (**12a–14a**), 2′-methoxy- (**16a–18a**) and challenging locked nucleic acids (**23a–24a** & **26a**). Interestingly C1 and base modifications are also well tolerated. PhoC_4 was also efficient at converting protected 3′-substituted nucleosides (**27a, 29a–30a**) that are important building blocks used in non-templated approaches to enzymatic oligonucleotide synthesis[12–15,19]. With a view to future process intensification and scale-up, we next explored the activity of PhoC_4 at elevated nucleoside concentrations (25–100 mM, Supplementary Table 1). Importantly even at low catalyst loadings (20 μM) and using only twofold excess of pyrophosphate, high reactions conversions are maintained across a range of modified substrates. These studies reveal that PhoC_4 can readily tolerate 10% DMSO (Supplementary Fig. 4) as a cosolvent and in favourable cases can achieve >5000 turnovers.

## An optimized platform for modified NTP synthesis

Next, our attention turned to the elaboration of PhoC-produced NMPs into the corresponding NTPs using a combination of PPKs and AcK. To this end, we produced a panel of 9 PPK homologs[39,40] and evaluated their activity towards a subset of NMPs, including those containing each of the four nucleobases, 3′-protecting groups and the most common 2′-modifications found in oligonucleotide therapeutics. These NMP substrates were prepared via PhoC_4 mediated biotransformations under intensified conditions followed by filtration to remove PhoC_4, precipitation of unreacted pyrophosphate using MgCl$_2$ and pH adjustment to the optimal operating window of PPK. From these assays, we were able to identify PPK homologs with activity towards all substrates tested to produce mixtures of NDP and NTP products (Fig. 4A and Supplementary Table 2). As an alternative to this two-step biotransformation PhoC_4 and PPK can be used in a single enzymatic cascade if pyrophosphate, which inhibits PPK activity, is replaced by phenyl phosphate as a donor (Supplementary Fig. 5). To further improve conversion to the target NTPs, we included a wild-type acetate kinase (AcK) from *Thermotoga maritima* that phosphorylates NDPs using acetyl phosphate as a donor[29]. With all substrates tested, addition of AcK resulted in substantially improved NTP conversion demonstrating the substrate promiscuity of the wild-type enzyme. Overall, using our three-enzyme system we successfully produced nine structurally diverse NTPs, featuring different nucleobases, 2′- and 3′-modifications, from the parent nucleoside in conversion ranging from 40 to 82% (Fig. 4B).

## Enzymatically produced NTPs in oligonucleotide synthesis

To demonstrate the practical utility of our biocatalytic approach, we next performed preparative scale transformations using 2′-MOE-ATP (**1d**) as a synthetic target. Under optimized conditions, the parent nucleoside (163 mg) was converted to 2′-MOE-ATP in 76% conversion (2′-MOE-ADP, 8%; 2′-MOE-AMP, 6%; 2′-MOE-A, 10%) using <0.06 mol% loading of each of the PhoC_4, PPK and AcK biocatalysts. Importantly, the synthesis required no costly intermediate purification steps. The reaction conversions are almost identical to those achieved in small-scale biotransformations, showing that our approach can be readily

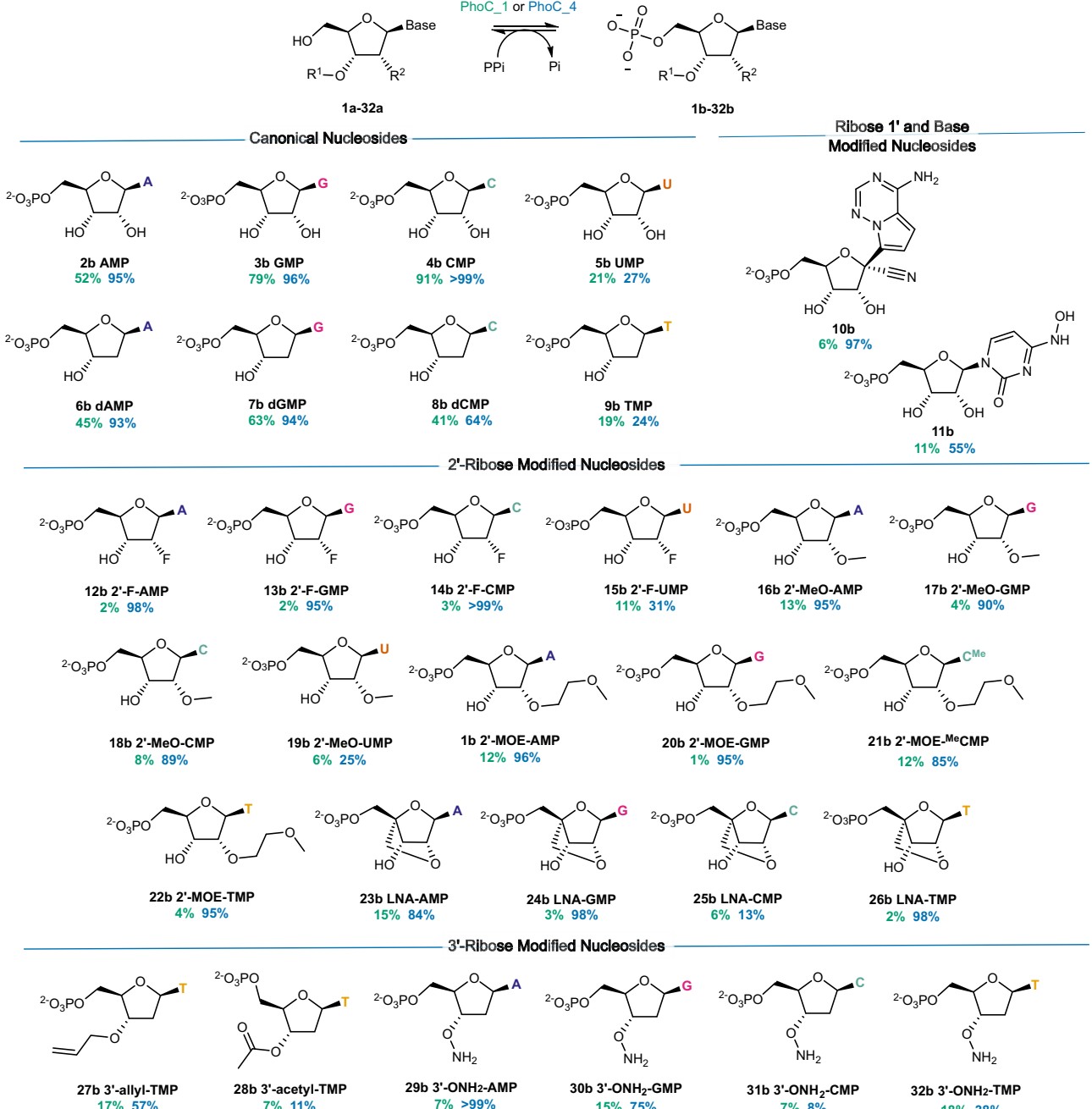

**Fig. 3 | Substrate scope of acid phosphatases PhoC_1 and PhoC_4.** The most highly evolved variant PhoC_4 (blue) provides NMP products (**1b**–**32b**) in higher conversion than the starting enzyme PhoC_1 (green). Biotransformations were performed using nucleoside (**1a**–**32a**, 1 mM), pyrophosphate (200 mM) and enzyme (20 μM) in 20 mM sodium acetate buffer pH 4 at 30 °C. A adenine, G guanine, C cytosine, U uracil, T thymine, ᴹᵉC 5-methylcytosine.

scaled from 100 μL to 10 mL without compromising efficiency. The 2′-MOE-ATP product was isolated following DEAE Sepharose ion exchange purification to provide **1d** in >99% purity and 65% isolated yield (Fig. 5A and Supplementary Fig. 6). Finally, we evaluated the performance of our enzymatically synthesized 2′-MOE-ATP (**1d**) as a substrate for biocatalytic oligonucleotide synthesis. Here, we employed our recently reported method that uses an engineered DNA polymerase and endonuclease V that work in synergy to amplify a catalytic self-priming template to furnish modified oligonucleotide sequences in a single operation (Fig. 5B)[18]. Using an engineered polymerase Tgo2M[46] and endonuclease V (EndoV) from *Thermatoga neopolitana*[18], enzymatically synthesized 2′-MOE-ATP was converted to a poly-2′-MOE-A 5mer oligonucleotide sequence. Importantly,

analogous reactions using chemically synthesized 2′-MOE-ATP purified by silica chromatography gave near identical results, affording the target sequence in >90% purity alongside truncated sequences as minor impurities (Fig. 5C). Thus, these results confirm the suitability of our enzymatically synthesized NTPs as building blocks for modified oligonucleotide production.

To further showcase the utility of our approach, we next elected to prepare 2′-F-ATP and investigate its incorporation into oligonucleotides. Given its close structural similarity to ATP, isolation of high-purity 2′-F-ATP from established ATP-dependent biocatalytic processes can be especially challenging. In this case, we prepared 2′-F-ATP from the parent nucleoside (713 mg) in 77% conversion (Supplementary Fig. 7). Seeking further improvements in efficiency, we recognized

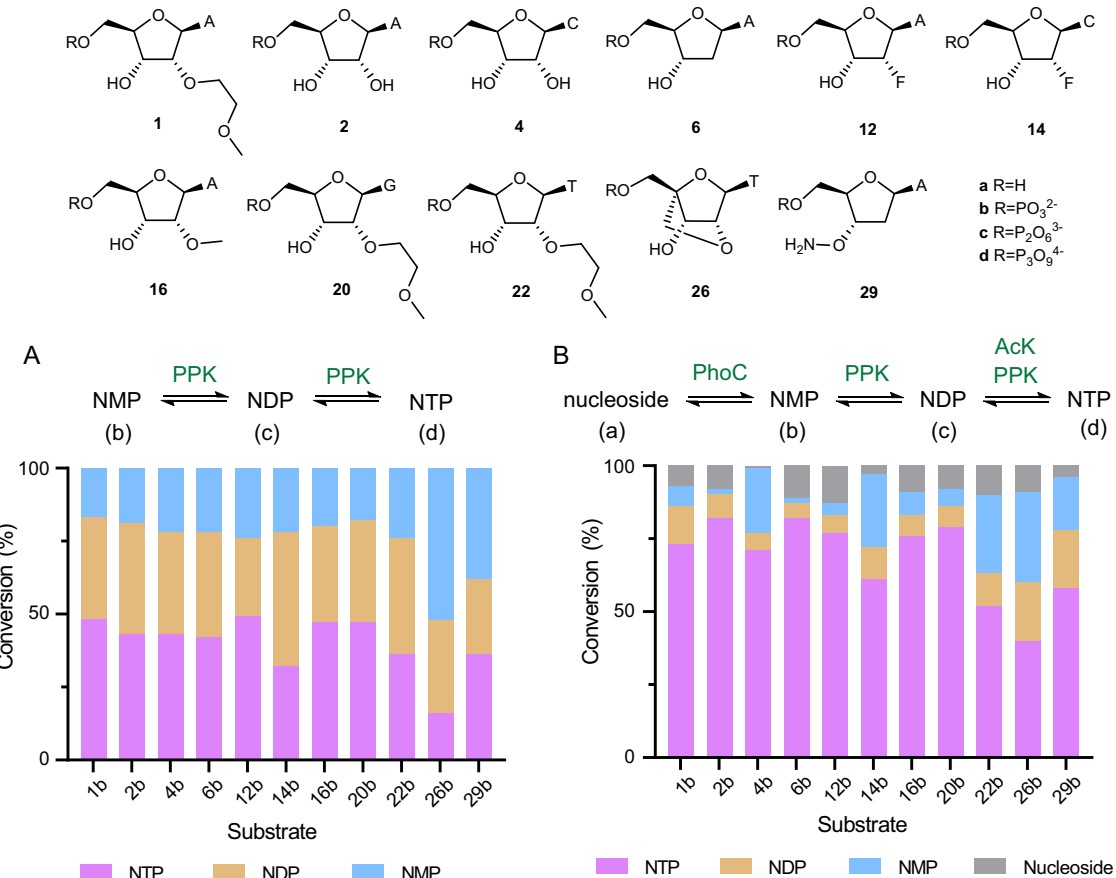

**Fig. 4 | An enzyme cascade for production of diverse modified NTPs. A** Bar chart showing the conversion of NMPs (blue) to NDP (orange) and NTP (magenta) mixtures using the most suitable PPK for each substrate (source data provided in Supplementary Table 2). **B** Bar chart showing the conversion of nucleoside substrates (grey) to NMP (blue) and NDP (orange) intermediates, and NTP (magenta) products using a combination of PhoC_4, PPKs, and AcKs (Source data are provided as a Source Data file).

the potential to use enzymatically produced 2′-F-ATP in crude form directly in biocatalytic oligonucleotide synthesis, thus obviating the need for NTP purification steps. To our delight, crude 2′-F-ATP could be used directly in our polymerase and endonuclease cascade to furnish a poly-2′-F-A 8mer in 70% conversion. Efforts to produce poly-2′-F-A sequences using 2′-F-ATP in the presence of low concentrations (from 1 to 10 mol%) of ATP contaminants gave complex product distributions arising from competing ATP and 2′-F-ATP incorporation (Supplementary Fig. 8).

We have established a sustainable and potentially scalable biocatalytic approach to NTP synthesis using inexpensive and widely available phosphate donors. Importantly, our method obviates the need for ATP, which poses numerous challenges for existing biocatalytic routes to NTP production. Our method is versatile and can be applied to the synthesis of a wide range of modified products containing 3′-protecting groups and 2′-modifications commonly found in therapeutic products. Key to the success of our platform was the development of an engineered acid phosphatase with broad substrate specificity. Moving forward, there are clear opportunities to make further improvements to our process. The development of highly evolved PhoC, PPK and AcK variants that are optimized towards target substrates of interest and minimize off-pathway processes (e.g., NMP hydrolysis by PhoC) will lead to enhanced productivity and reduced enzyme loadings. Here, there is also the potential to develop enzymes with similar pH-optima to avoid the need for pH adjustments during the process. For large scale NTP manufacture, product isolation and purification will preferably be carried out by precipitation and/or crystallization to avoid the need for ion exchange resins[29].

Alternatively, as shown within this study, crude NTPs from our process can be used directly in enzymatic oligonucleotide synthesis, avoiding the need for NTP purification. It is unlikely that such an approach could be adopted with ATP-dependent methods, as even low concentrations of ATP contaminants severely compromise the purity of downstream oligonucleotide products. Our approach also offers several advantages over established chemical methods for NTP synthesis that rely on the use of toxic and difficult to remove solvents such as trimethylphosphate, air/moisture-sensitive and counterion-specific reagents, and the preparation and isolation of reactive intermediates [P(V) or P(III) species]. In light of these advantages, we anticipate that our biocatalytic platform will be widely adopted for the preparation of canonical and modified NTPs needed to underpin the biocatalytic production of important RNA therapeutics[12–19].

## Methods
### Materials
Reagents and chemicals were obtained from commercial suppliers and were used without further purification unless otherwise stated. Lysozyme, kanamycin sulphate, tetrapotassium pyrophosphate, sodium hexametaphosphate, lithium potassium acetyl phosphate, **2c** and **2d** were purchased from Sigma-Aldrich, polymyxin B sulfate, **12a**, and **15a** from AlfaAesar, LB agar, LB medium, and Isopropyl β-D-1-thiogalactopyranoside (IPTG) from Formedium. Phenylphosphate, **2b**, **13a**, **16a**, **17a**, and **19a** were purchased from ThermoScientific, **2a**, **4a–8a**, and **18a** from Fluorochem, **3a** and **9a** from Tokyo Chemical Industry, **1a**, **14a**, **20a–22a** from Biosynth, **10a** and **11a** from MedChemExpress, **23a–26a** from Sapala Organics. Compounds **29a–32a**

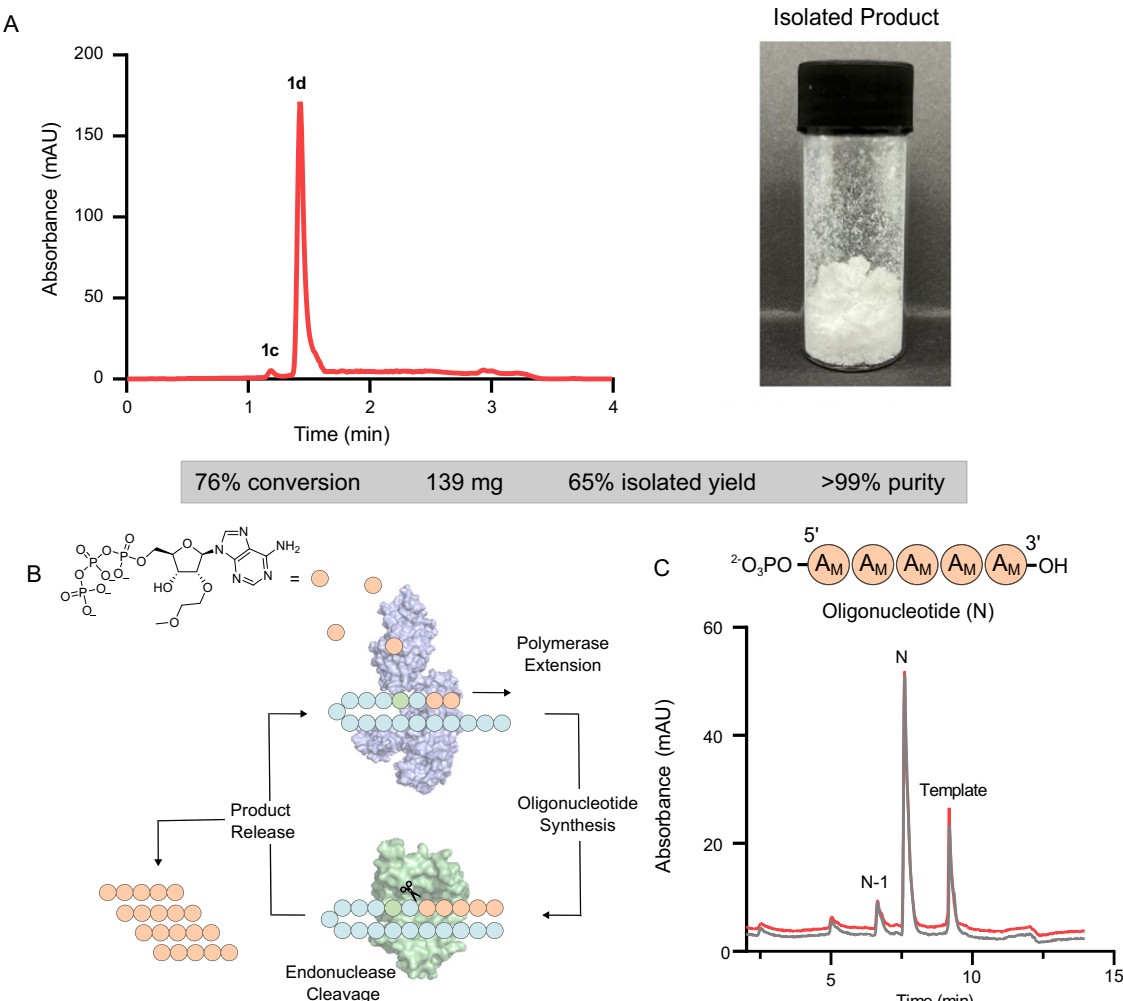

**Fig. 5 | 2′-MOE-ATP and 2′-MOE modified oligonucleotide synthesis.**
**A** Preparative scale synthesis of 2′-MOE-ATP (**1d**). HPLC trace and image of the isolated product. The minor impurity is 2′-MOE-ADP (**1c**); **B** A template dependent

enzyme cascade for the synthesis of oligonucleotides[18]. **C** HPLC traces of a poly-2′-MOE-A 5mer (N) synthesized using enzymatically (red) and chemically (grey) produced 2′-MOE-ATPs. Minor impurities include an N-1 truncated sequence.

were provided by Nuclera Ltd (Cambridge). Nucleosides **27a** and **28a**, standards of **1b** and **1d** were chemically synthesized in house (see supplementary information). *E. coli* DH5α, *E. coli* BL21 (DE3), Q5 DNA polymerase, T4 DNA ligase and restriction enzyme BsaI were purchased from New England BioLabs and *E. coli* C43 (DE3) from Cambridge Bioscience. Oligonucleotides were synthesized by Integrated DNA Technologies. PPK4-PPK9 were obtained from Prozomix. Diafiltration devices (Vivaspin) were purchased from Sartorius AG. Sequencing was performed by Source BioScience (Nottingham). Expression plasmids: Genes encoding PhoC_1, PPK1, PPK2, PPK3, and AcK were ordered from Integrated DNA Technologies (IDT) as gBlocks with codon optimization for recombinant expression in *E. coli*. The genes were cloned into pET-28a using Golden Gate Assembly (Nucleotide sequences of ORFs are shown below).

### Protein expression and purification

For recombinant expression of PPKs and AcK, the respective plasmids were transformed into chemically competent *E. coli* BL21 (DE3) cells. For expression of PhoC and variants, chemically competent *E. coli* C43 (DE3) were transformed with the relevant pET28a_PhoC constructs. Following transformation and plating on LB agar containing 50 μg/mL kanamycin, 5 mL starter cultures of LB medium (50 μg/mL kanamycin) were inoculated with single colonies of freshly transformed cells and incubated for 18 h at 37 °C and 180 rpm. Starter cultures were used to

inoculate 50 mL of LB medium supplemented with 50 μg/mL kanamycin. Cultures were grown at 37 °C, 180 rpm to an optical density of ~0.6 at 600 nm (OD$_{600}$). Protein expression was induced with the addition of IPTG to a final concentration of 1 mM. Induced cultures were incubated for 20 h (PPKs and AcK) or 4 h (PhoC and variants) at 30 °C, 180 rpm. Cells were subsequently harvested by centrifugation (3220 × *g* for 10 min). Pelleted cells were resuspended in lysis buffer (PhoC and variants: 1 mg/mL lysozyme, 0.5 mg/mL polymyxin B, 20 mM sodium acetate, 300 mM NaCl, pH 7.5; PPKs & AcK: 1 mg/mL lysozyme, 0.5 mg/mL polymyxin B, 50 mM Tris-HCl, 300 mM NaCl, pH 7.5) and lysed by sonication. Cell lysates were cleared by centrifugation (27,216 × *g* for 30 min). Supernatants were loaded onto Ni-NTA agarose (Qiagen) gravity-flow columns for affinity purification and washed with wash buffer (PhoC and variants: 20 mM sodium acetate, 300 mM NaCl, pH 7.5, 10 mM imidazole; PPKs & AcK: 50 mM Tris-HCl, 300 mM NaCl, 10 mM imidazole, pH 7.5). Proteins were eluted using 500 mM imidazole, 300 mM NaCl, 20 mM sodium acetate, pH 7.5 for PhoC and variants, or 50 mM Tris-HCl, pH 7.5 for PPKs & AcK. Proteins were desalted on 10DG gel filtration columns (Bio-Rad) equilibrated with buffer (PhoC and variants: 20 mM sodium acetate, pH 5.0; PPKs & AcK: 50 mM Tris-HCl, pH 7.5) and analysed by SDS−PAGE. Proteins were aliquoted, flash-frozen in liquid nitrogen and stored at −80 °C. Protein concentrations were determined by measuring the absorbance at 280 nm on a NanoDrop spectrometer (Nanodrop 1000.3.8.1). Extinction

coefficients were calculated based on the respective amino acid sequence using Expasy ProtParam (https://web.expasy.org/protparam/).

### PhoC library generation

Selected amino acid positions were individually randomized using degenerate NNK primers (see supplementary Table S3). DNA libraries were generated by overlap extension polymerase chain reaction (PCR). Assembled and gel purified genes were subsequently inserted into a customised pET 28a vector using Golden Gate Assembly in a 5:1 ratio, respectively, using T4 DNA ligase (20 U/μL) and BsaI (1 U/μL). Golden gate reactions were directly transformed into *E. coli* DH5α cells. The resulting colonies on LB-agar (50 μg/ml kanamycin) plates were pooled together, and plasmids were extracted by Miniprep (QIAGEN) to yield plasmid DNA for each library.

### PhoC library screening

For protein expression and screening, all transfer and aliquoting steps were performed using Hamilton liquid-handling robots running Alvaro Cuevas 2010 software. Chemically competent *E. coli* C43 (DE3) cells were transformed with the appropriate library plasmids and plated on LB agar (50 μg/ml kanamycin) plates. Freshly transformed colonies were used to inoculate 180 μL of LB (50 μg/mL kanamycin) in Corning Costar 96-well microtitre round-bottom plates. Each plate contained six freshly transformed clones of the parent template PhoC_1 and two catalytically inactive clones (PhoC_1_H168A/H207A) in *E. coli* C43 (DE3). Plates were sealed with gas permeable membranes (Wolflabs) and incubated overnight at 30 °C, 80% humidity in a shaking incubator at 850 rpm. 20 μL of overnight culture was used to inoculate 480 μL of LB medium supplemented with 50 μg/mL kanamycin in 2 mL 96 deep well plates. The cultures were incubated at 30 °C, 80% humidity with shaking at 850 rpm until an $OD_{600}$ of -0.6 was reached. Protein expression was induced by the addition of IPTG to a final concentration of 1 mM. Induced plates were incubated for 4 h at 30 °C, 80% humidity, with shaking at 850 rpm. Cells were harvested by centrifugation at $3220 \times g$ for 10 min. The supernatant was discarded and the cell pellets were resuspended in 200 μL of lysis buffer (20 mM sodium acetate, 1 mg/mL lysozyme, and 0.5 mg/mL polymixin B) and incubated for 2 h at 30 °C, 80% humidity with shaking at 850 rpm. Cell debris was removed by centrifugation at $3220 \times g$ for 10 min, and 50 μL of cell free extract was transferred to fresh 96-well microtitre plates. 50 μL Reaction mixture was added to give a final concentration of 200 mM potassium pyrophosphate, 1 mM 2′-MOE-adenosine (**1a**), and 20 mM sodium acetate (pH 4). Plates were incubated at 30 °C, 850 rpm, and 80% humidity and after 16 h, the reaction was terminated by the addition of 100 μL methanol (final concentration 50% v/v). After 10 min centrifugation at $3220 \times g$, 50 μL of the reaction mixture was transferred to fresh 96-well microtitre plates for UPLC analysis. Plasmids from variants with improved activity were isolated and sequenced. The most active clones from each round were rescreened as purified proteins. Following each round of evolution, beneficial mutations were combined by overlap extension PCR and evaluated as purified proteins. The best variant was selected as the parent template for the subsequent round of directed evolution.

### Chromatographic analysis

LC-MS analysis was performed using an Agilent 1290 Infinity II LC/MSD running in negative ESI mode at 4.0 kV capillary, acquiring up to 2000 m/z. Signals were recorded at 260 nm and analysed using Agilent OpenLab software (3.6). Nucleosides and NMPs were analysed using an InfinityLab Poroshell 120 EC-C18 column (2.1 × 100 mm, 1.9 μm). Compounds were eluted over 10 min using a gradient of 5–70% eluent B at 0.6 mL/min. Eluent A: 10 mM ammonium acetate, pH 3.6; Eluent B: Acetonitrile. Oligonucleotides were analysed on an AdvanceBio Oligonucleotides 2.7 μm column, 50 × 2.1 mm (Agilent) at 60 °C.

Oligonucleotide sequences were separated over 14 min using a gradient of 2–50% eluent B at 0.3 mL/min. Eluent A: 400 mM hexafluoro isopropanol and 20 mM triethylamine in water; Eluent B: Methanol.

All HPLC analysis was performed on an Agilent 1290 Infinity II LC system, signals were recorded at 260 nm. Peak areas were integrated using Agilent OpenLab software (3.6) and assigned by comparison to commercial or chemically synthesized standards. PhoC libraries were evaluated using an InfinityLab Poroshell 120 EC-C18 column (3.0 × 30 mm, 2.7 μm). Eluent A: 60 mM ammonium acetate, pH 3.6; Eluent B: Acetonitrile. Separation of 2′-MOE-adenosine monophosphate (**1b**) from 2′-MOE-adenosine (**1a**) was achieved using a gradient of 5–12% B at 1 mL/min for 1 min. PPK and AcK reactions were analysed using a ZORBAX Eclipse Plus C18 column (2.1 × 50 mm, 1.8 μm) at 40 °C. Eluent A: 50 mM hydroxylamine adjusted to pH 7 with acetic acid; Eluent B: 65:35 (v/v) mixture of isopropanol:acetonitrile. Nucleosides and nucleotides (NMP, NDP, NTP) were eluted over 4 min using a gradient of 5–90% eluent B at 0.6 mL/min.

### Steady state kinetic assays

Kinetic assays were performed using pyrophosphate (200 mM), 2′-MOE-A (**1a**, 0–256 mM) and enzyme (5 μM) in 20 mM sodium acetate buffer pH 4 at 30 °C. Samples taken at different time points were quenched with the addition of 1 volume methanol. Initial rates were measured using HPLC and product concentration was determined by comparison to commercial standards. Assays were performed in triplicate, and the averaged initial rates were fitted to the Michaelis–Menten equation using GraphPad Prism 9 software (9.1.2) (Supplementary Fig. 2).

### Substrate profiling of PhoC_1 and PhoC_4

100 μL reactions containing 1 mM nucleoside (**1–32a**), 200 mM pyrophosphate, 20 μM purified PhoC_1 or PhoC_4, 20 mM sodium acetate, pH 4 were assembled and incubated at 30 °C, 850 rpm. Reactions containing no enzyme served as a negative control. Reactions were quenched with the addition of 100 μL methanol and precipitated protein was removed by centrifugation. Cleared reactions were analysed by LC-MS analysis. In total 96 samples were evaluated, using a single replicate.

### Evaluation of PhoC_4 R201A and H168A variants

Arg201 and His168 were individually mutated to alanine in PhoC_4. DNA genes were generated by overlap extension PCR. Assembled and gel purified genes were subsequently inserted into a customised pET 28a vector using Golden Gate Assembly in a 5:1 ratio, using T4 DNA ligase (20 U/μL), and BsaI (1 U/μL). Golden gate reactions were directly transformed into *E. coli* DH5α cells. Plasmids were extracted by Miniprep (QIAGEN) and mutations were confirmed by sequencing. PhoC_4 R201A and PhoC_4 H168A were expressed and purified according to the general protocol. To evaluate their activity, reactions containing 1 mM 2′-MOE-A, 200 mM pyrophosphate, 20 μM purified PhoC_4, PhoC_4 R201A or PhoC_4 H168A, 20 mM sodium acetate, pH 4 were assembled and incubated at 30 °C, 850 rpm. Reactions were quenched with the addition of 100 μL methanol and precipitated protein was removed by centrifugation. Cleared reactions were analysed by LC-MS analysis and the conversion to product was determined by comparison to chemically synthesized 2′-MOE-AMP (see supplementary information).

### Substrate profiling of a PPK panel

1 mL reactions containing 25–100 mM nucleoside (Supplementary Table 1), 200 mM potassium pyrophosphate, 20 μM purified PhoC_4, 20 mM sodium acetate, pH 4 were assembled and incubated at 30 °C, 850 rpm for 4–16 h. Next, PhoC was removed by diafiltration using 10 kDa filter to terminate the reaction. The filtrate was collected and pH adjusted to 7 by addition of 400 μL 200 mM Tris-HCl, pH 9 and

200 μL 5 M NaOH. Residual pyrophosphate was precipitated by addition of 400 μL 750 mM MgCl$_2$. After centrifugation, 50 μL of the buffer adjusted NMP containing solution was mixed with 50 μL reaction mixture. The final reaction contained 20 mg/mL sodium hexametaphosphate, 50 mM MgCl$_2$, 1 mg/mL lyophilized PPK cell free extract (Prozomix), 50 mM Tris-HCl, pH 7 and was incubated at 30 °C, 850 rpm for 16 h. Reactions were terminated by diafiltration (10 kDa filter) and analysed by HPLC.

### One-pot PhoC_4 and PPK cascade
A 100 μL reaction containing 10 mM adenosine (**2a**), 200 mM potassium pyrophosphate or phenylphosphate, 20 mg/mL sodium hexametaphosphate, 50 mM MgCl$_2$, 20 μM purified PhoC_4, PPK5 (Prozomix) as lyophilized cell-free extracts (1 mg/mL), 20 mM sodium acetate and 50 mM Tris-HCl pH 5 were assembled and incubated at 30 °C and 850 rpm for 16 h. Reactions were terminated by diafiltration (10 kDa filter) and analysed by HPLC.

### Analytical scale NTP synthesis
1 mL reactions containing 25–100 mM nucleoside, 200 mM potassium pyrophosphate, 20 μM purified PhoC_4 and 20 mM sodium acetate pH 4 were assembled and incubated at 30 °C, 850 rpm for 4–16 h (Supplementary Table 1). Next, PhoC_4 was removed by diafiltration using a 10 kDa filter to terminate the reaction. The filtrate was collected and pH adjusted to 7 by addition of 400 μL 200 mM Tris-HCl pH 9 and 200 μL 5 M NaOH. Residual pyrophosphate was precipitated by addition of 400 μL 750 mM MgCl$_2$. After centrifugation, 50 μL of the buffer-adjusted NMP solution was mixed with 50 μL reaction mixture. The final reaction contained 20 mg/mL sodium hexametaphosphate, 50 mM MgCl$_2$, 1 mg/mL lyophilized PPK cell free extract (Prozomix) and 50 mM Tris-HCl pH 7, and was incubated at 30 °C, 850 rpm for 16 h. The reaction was terminated by diafiltration (10 kDa filter) and 50 μL of filtrate was added to 50 μL AcK reaction mixture. The final reaction contained 50 mM lithium potassium acetylphosphate, 5 mM MgCl$_2$, 25 μM purified AcK, and 50 mM Tris-HCl pH 7, and was incubated at 30 °C, 850 rpm for 16 h. Reactions were terminated by diafiltration (10 kDa filter) and analysed by HPLC.

### Preparative scale 2′-MOE-ATP (1d) synthesis
All incubation steps were performed using an EasyMax 102 Advanced Thermostat reactor (Mettler Toledo) equipped with a magnetic stir bar and incubated at 30 °C, 100 rpm. A 10 mL reaction was prepared containing 50 mM 2′-MOE-adenosine (**1a**), 200 mM potassium pyrophosphate, 20 μM purified PhoC_4 and 20 mM sodium acetate, pH 4. Following 12 h incubation PhoC_4 was removed by diafiltration using a 10 kDa filter to terminate the reaction. The filtrate containing 2′-MOE-AMP (**1b**) was collected and pH adjusted to 7 by adding 4 mL 200 mM Tris-HCl, pH 9 and 2 M 5 M NaOH. Residual pyrophosphate was removed by precipitation with 4 mL 750 mM MgCl$_2$. Following centrifugation, the 2′-MOE-AMP (**1b**) containing supernatant was added to a 20 mL reaction mixture containing 20 mg/mL sodium hexametaphosphate, 50 mM MgCl$_2$, 30 μM purified PPK2 and 50 mM Tris-HCl pH 7. Following a further 16 h incubation, the reaction was terminated by diafiltration (10 kDa filter), and the filtrate was added to a 25 mL final reaction mixture containing 50 mM lithium potassium acetylphosphate, 5 mM MgCl$_2$, 25 μM purified AcK, and 50 mM Tris-HCl, pH 7. After 16 h AcK was removed by diafiltration (10 kDa filter).

### Purification and isolation of 2′-MOE-ATP (1d)
2′-MOE-ATP (**1d**) was purified from the crude reaction mixture by DEAE sepharose ion exchange chromatography on an ÄKTA pure™ (running Cytiva Unicorn software 3.6), equipped with a Cytiva GE XK 26 column packed with DEAE Sepharose Fast Flow resin. Signals were recorded at 260 nm. Eluent A: water; Eluent B: 1 M ammonium bicarbonate. Separation of **1d** from **1a**, **1b** and **1c** was achieved using a gradient of

15–100% B. The purified product was isolated by freeze drying overnight and stored at −80 °C (Supplementary Fig. 4). $^1$H NMR (400 MHz, D$_2$O) δ 8.51 (s, 1H), 8.25 (s, 1H), 6.19 (d, $J$ = 6.2 Hz, 1H), 4.68 (dd, $J$ = 5.2, 3.1 Hz, 1H), 4.59 (t, $J$ = 5.7 Hz, 1H), 4.45–4.35 (m, 1H), 4.31–4.13 (m, 2H), 3.83 (ddd, $J$ = 11.8, 5.8, 3.2 Hz, 1H), 3.70 (ddd, $J$ = 11.9, 5.4, 3.2 Hz, 1H), 3.57–3.42 (m, 2H), 3.14 (s, 3H) ppm; $^{31}$P{$^1$H} NMR (162 MHz, D$_2$O) δ −7.68 (br s), −11.20 (d, $J$ = 19.4 Hz), −22.30 (t, $J$ = 19.4 Hz).

### Poly-2′-MOE-A 5mer biosynthesis
A solution containing 4 μM DNA template (5′- (Bio)TTTTTCACGTG-CACCATTGGTGCACGIG) and 1 mM **1d** in buffer (50 mM Tris-AcOH, pH 8, 20 mM MgSO$_4$, 100 mM potassium glutamate, 50 mM KOAc, 0.01 mg/ml Acetylated BSA, 30 mM DTT) was prepared according to previous reports[18]. The reaction was initiated with the addition of 5 μM DNA polymerase Tgo2M[46] and 0.3 μM TnEndoV[18] and incubated at 70 °C for 12 h. Reactions were quenched with the addition of one volume 20 mM EDTA. Samples were heated at 98 °C for 2 h and precipitated proteins were removed by centrifugation (14,000 × g) for 15 min. The supernatant was analysed by LC-MS.

### Preparative scale 2′-F-ATP (12d) synthesis
All incubation steps were performed using an EasyMax 102 Advanced Thermostat reactor (Mettler Toledo) equipped with a magnetic stir bar and incubated at 30 °C, 100 rpm. A 50 mL reaction was prepared containing 50 mM 2′-F-adenosine (**12a**), 200 mM potassium pyrophosphate, 20 μM purified PhoC_4 and 20 mM sodium acetate, pH 4. Following 12 h incubation, PhoC_4 was removed by diafiltration using a 10 kDa filter to terminate the reaction. The filtrate containing 2′-F-AMP (**12b**) was collected and pH adjusted to 7 by adding 20 mL 200 mM Tris-HCl, pH 9 and 10 mL 5 M NaOH. Residual pyrophosphate was removed by precipitation with 20 mL 750 mM MgCl$_2$. Following centrifugation, the 2′-F-AMP (**12b**) containing supernatant was added to a 150 mL reaction mixture containing 20 mg/mL sodium hexametaphosphate, 50 mM MgCl$_2$, 30 μM purified PPK3 and 50 mM Tris-HCl pH 7. Following a further 16 h incubation, the reaction was terminated by diafiltration (10 kDa filter), and the filtrate was added to a 200 mL final reaction mixture containing 50 mM lithium potassium acetylphosphate, 5 mM MgCl$_2$, 25 μM purified AcK, and 50 mM Tris-HCl, pH 7. After 16 h AcK was removed by diafiltration (10 kDa filter) and the filtrate containing 2′-F-ATP (**12d**) was collected. The 2′-F-ATP concentration was measured using HPLC analysis by comparison to a commercial standard.

### Poly-2′-F-A 8mer biosynthesis
A solution containing 4 μM DNA template (5′- (Bio) TTTTTTTTTCACGTGCACCATTGGTGCACGIG), 10% formamide, 10 mM (NH$_4$)$_2$SO$_4$, 10 mM KCl, 2 mM MgSO$_4$, 0.1% Triton ×-100, 2 mM NiCl$_2$ and 20 mM Tris-HCl pH 8.8 was prepared according to previous reports[18]. Enzymatically synthesized 2′-F-ATP was added to give a final concentration of 1 mM. The reaction was initiated with the addition of 1 μM DNA polymerase TfPol*, 2 μM TnEndoV, and 4.5 U/mL pyrophosphatase, and incubated at 70 °C for 12 h. Reactions were quenched with the addition of one volume of chloroform. After centrifugation, the supernatant was analysed by LC-MS.

### PhoC crystallization and modelling
Single crystals of PhoC_1 and PhoC_4 were prepared by mixing 200 nl of 10 mg ml$^{-1}$ protein in crystallization buffer with equal volumes of precipitant. All trials were conducted by sitting drop vapour diffusion and incubated at 4 °C. Crystals of PhoC_1 appeared from a precipitant of 0.1 M Phosphate/Citrate pH 5.5 and 30% PEG Smear Low (12.5% v/v PEG 400, 12.5% v/v PEG 500 MME, 12.5% v/v PEG 600 and 12.5% PEG 1000) (BCS screen A2). PhoC_4 crystals were grown from a precipitant solution containing 0.1 M Carboxylic acids (0.2 M Sodium formate, 0.2 M Ammonium acetate, 0.2 M Sodium citrate tribasic dihydrate, 0.2 M Potassium sodium tartrate tetrahydrate and 0.2 M Sodium

oxamate), 0.1 M Buffer system 3 pH 8.5 (1.0 M Tris base, Bicine pH 8.5), 37.5% v/v Precipitant Mix 4 (25% v/v MPD, 25% PEG 1000, 25% PEG 3350) (Morpheus G12). Crystals were flash cooled in liquid nitrogen prior to data collection. Data were collected from single crystals of PhoC_1 and PhoC_4 at Diamond Light Source (MX31850) and subsequently scaled and reduced with Xia2[47]. Preliminary phasing was performed by molecular replacement using an AlphaFold 2 derived search model. Iterative cycles of rebuilding and refinement were performed in COOT[48] and Phenix.refine[49], respectively. Structure validation with MolProbity[50] and PDBREDO[51,52] was integrated into the iterative rebuild and refinement process. The resolution cut of the data was determined by paired refinement as implemented in PDBREDO. Complete data collection and refinement statistics can be found in Table S4. Coordinates and structure factors were deposited in the Protein Data Bank (PDB) under accession codes 9QCI (PhoC_1) and 9 QCJ (PhoC_4). Representative electron density maps for PhoC_1 and PhoC_4 are shown in Supplementary Fig. S9. Docking studies were performed in ICM-Pro (Molsoft) utilising a model derived from the crystal structure of PhoC_4. In the absence of a substrate or product bound complex of PhoC_4, it was necessary to generate a model incorporating features observed in 1EOI (the crystal structure of *Escherichia blattae* acid phosphatase complexed with the transition state analogue molybdate)[45]. The conformation of 1EOI was imposed onto the coordinates of PhoC_4 using molecular mechanics. All subsequent docking studies were directed towards this derived model of PhoC_4*.

### Statistics and reproducibility

No statistical method was used to predetermine sample sizes. No data were excluded from the analyses. The experiments were not randomized and the investigators were not blinded to allocation during experiments and outcome assessment.

### Reporting summary

Further information on research design is available in the Nature Portfolio Reporting Summary linked to this article.

## Data availability

The data supporting the findings of this study are available within this Article and its Supplementary Information. Coordinates and structure factors for PhoC_1 and PhoC_4 were deposited in the Protein Data Bank (PDB) under accession codes 9QCI and 9QCJ, respectively. Source data are provided with this paper.

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

## Acknowledgements

The authors are very grateful to Nuclera for the gift of 3′-O-NH$_2$ nucleosides **29–32b**. The authors are grateful for generous funding from the UK Research and Innovation Council (Future Leader Fellowship MR/T041722/1 to S.L.L.), Medical Research Council (Nucleic Acid Therapy Accelerator Grant MR/W029324/1 to Q.M., C.M., C.B., A.N.C., S.D., M.W., P.S.B., G.M., N.J.T., S.L.L.) and the Medicines Manufacturing Innovation Centre (S.L.L.).

## Author contributions

S.L.L., N.J.T. and G.M. designed the project. Q.M., C.B., C.M., Y.Z., A.E., A.N.C., S.D., R.O., M.O., C.L., J.D.F., and S.J.C. performed the experiments and collected the data. Q.M. and S.L.L. wrote the manuscript.

## Competing interests

S.L.L. and Q.M. have filed a patent application to the UK Patent office (application number GB 2517296.6) describing our biocatalytic method of NTP synthesis and engineered PhoC sequences. S.L.L. has filed an international patent describing the application of a polymerase and endonuclease V cascade for production of therapeutic oligonucleotides (WO2023/041931). The remaining authors declare no competing interests.
