## [Transparent Peer Review file · Nature Communications]

Enzymatic Synthesis of Key RNA Therapeutic Building Blocks Using Simple Phosphate Donors

Corresponding Author: Dr Sarah Lovelock

Version 0:

Reviewer comments:

Reviewer #1

(Remarks to the Author)

This manuscript describes an ATP-free biocatalytic approach that can be utilized to convert nucleosides of diverse chemistry into their corresponding nucleoside triphosphates (NTPs). To start the pipeline, the authors engineered a non-specific acid phosphatase (PhoC) to synthesize nucleoside monophosphates (NMPs) using inorganic pyrophosphatase as the phosphate donor. The most promising variant (PhoC_4), obtained by combing beneficial mutations identified by screening variants in microtiter plates, achieves 96% conversion of 2'-MOE-A into 2'-MOE-AMP. Structure determination studies and molecular modeling of PhoC_4 suggest that the flexible loop plays an important role in substrate recognition. Importantly, PhoC_4 exhibits improved activity across a range of nucleosides relative to the starting variant (PhoC_1). An optimized enzymatic cascade was then established utilizing PPK and AcK to convert NMPs into NTPs, with the successful demonstration of nine structurally diverse examples from their starting nucleoside. The enzymatic pathway was scaled, under optimized conditions to generate >150 mg of 2'-MOE-ATP, which was used to produce a poly-2'-MOE-oligonucleotide pentamer. Overall, the manuscript is interesting, but will require significant revision prior to publication.

Comments:

1. The main concern is that certain areas of the manuscript are underdeveloped, lacking appropriate context, discussion, and referencing. Other areas contain jargony terms that should be avoided.
2. Important details in the results and discussion section are lacking. Some examples include: (i) undefined statements such as "high conversion" on line 75; (ii) theoretical size of the library; (iii) number of clones evaluated; and (iv) a clear description of the model in Figure 2E as a chimera from prior structures.
3. Nonstandard terms should be removed from the text. This includes "holistic biocatalytic approach", "telescoped fashion", "vide infra", and "pleasingly".
4. The manuscript lacks adequate citations. Some examples include: (i) the MOE chemical modification (line 95); (ii) the synthesis of 100 g of 5' IMP (line 86); and the entire methods paragraph on the crystal structure which fails to cite any of the programs used for structure refinement.
5. The source of compound 1a is missing. Was this synthesized or purchased? If it was synthesized, the details are needed.
6. The synthesis of a pentamer of 2'-MOE-modified polyA is not a suitable demonstration of an oligonucleotide therapeutic. The language around this example should be tempered. Discussion about the length versus T_m values of the extended product is warranted.
7. The x-ray crystal structures of PhoC_1 and PhoC_4 warrant a more detailed structural comparison.
8. The functional role of H168 and R183 should be empirically determined.
9. The structural consequences of adaptive PhoC mutations should be discussed and shown in a figure, in particular E90, which is not located in the flexible loop.

10. Molybdate is mentioned in the text, but not in the structure.

11. In certain places, figure callouts are unclear. For example, the substrates from Figures 2 and 3 are being called out as 1a, 1b, etc. It is not clear which figure they are referring to.

12. The comparison between the enzymatically produced poly-2'-MOE-A 5-mer and the authentic standard in Figure 5c should be more clearly described in the text.

13. The manuscript would benefit from a summary of the strengths and weaknesses of the current strategy relative to other NTP synthesis approaches. Yields, time, purification, solvent, etc. would be useful information to share.

Reviewer #2

(Remarks to the Author)

The manuscript by Meng et al. reports the development of a fully enzymatic, ATP-free cascade for the production of nucleoside triphosphates (NTPs), including chemically modified variants used in therapeutic oligonucleotides. This work elegantly builds on the seminal work in Science (DOI: 10.1126/science.add589), also by the Lovelock group, to address a key bottleneck in the RNA therapeutics field, namely, the scalable and selective synthesis of modified NTPs, which are often challenging to prepare chemically. The work describes a three-enzyme system comprising an engineered acid phosphatase (PhoC), polyphosphate kinase (PPK), and acetate kinase (Ack), which together enable phosphorylation of a wide array of canonical and modified nucleosides using inexpensive phosphate donors and without the need for ATP. This approach significantly simplifies purification by eliminating ATP and ADP impurities that would otherwise complicate downstream processes.

The authors demonstrate a well-executed protein engineering campaign on PhoC to enhance its activity on therapeutically relevant substrates such as 2'-MOE-adenosine. Directed evolution led to a triple mutant (PhoC_4) with an 8-fold improvement in conversion and an expanded substrate scope to several therapeutically relevant compounds. Structural characterization of PhoC_4 and docking of 2'-MOE-AMP provide insight into the mutational effects and highlight a loop region as likely contributing to substrate engagement and activity partitioning. This sets the stage for future engineering efforts and adds mechanistic depth to the study. Following this, a panel of PPKs and a promiscuous Ack were used to advance monophosphates to the triphosphate level across a range of substrates, achieving conversions up to 82%. The enzymatic synthesis of 2'-MOE-ATP on preparative scale and its subsequent use in oligonucleotide assembly via a polymerase–endonuclease cascade further demonstrates the practical value of the method.

The manuscript is clearly written, methodically executed, and of broad relevance. Overall, this is an excellent contribution to the field and represents a clear advance in enabling the biocatalytic synthesis of NTP building blocks. The work will be of interest to researchers in biocatalysis, nucleic acid chemistry, and RNA therapeutic development. As such I recommend publication after minor revisions below:

- While the engineering of PhoC_4 is thoroughly presented in terms of conversion data and structural interpretation, a kinetic comparison between PhoC_1 and PhoC_4 (e.g., kcat, Km values) would provide more quantitative context to the improvements and help evaluate trade-offs between rate and specificity.

- The preparative scale synthesis is a great addition to the manuscript. It may also be helpful to include a brief comment on how the system behaves at larger scales - any limitations observed with respect to enzyme stability, precipitation issues, solubility, for example.

Reviewer #3

(Remarks to the Author)

Meng et al report on the phosphorylation of nucleosides by enzymatic cascade reactions. The resulting nucleoside triphosphates are substrates of engineered RNA polymerases in the synthesis of oligonucleotides. The authors use transphosphorylation catalyzed by an engineered phosphatase to produce the nucleoside monophosphates (NMP). The NMPs are phosphorylated sequentially to yield the nucleoside triphosphates (NTP). The nucleoside diphosphates (NDP) are formed as intermediates. The phosphorylations shown can be done chemically or through enzymatic cascade reactions, typically using NTP dependent kinases. The authors use enzymatic reactions independent of NTP and claim that NTP-free cascade transformations represent a major advance. This reviewer did not agree. The study is sound but the results do not constitute a significant improvement to existing methods of synthesis. The enzymatic reactions are all well-known and the developments are rather incremental. Most importantly, complicated mixtures of nucleoside, NMP, NDP and NTP are produced (Figure 4) that require extensive purification before the NTP product can be used in further transformations. The argument used against the use of established NTP kinases, that the intermediate donor (e.g., ATP) must be removed as part of the product work-up, appears to be biased. The requirement for work-up of mixtures from reactions as shown in Figure 4 certainly exceeds the purification needed for a classical NTP-kinase cascade reaction.

Figure 1 is biased in the description of the different methods. While the reviewer agrees on some limitations of the chemical methods, these reactions aren't as low yielding and complex as the authors would like to suggest. The distinction made between reactions in B and C are not clear and appear to reflect the subjective preferences of the authors. The effort in product purification required in each of these reactions depends on the specific transformation used and the extent to which the particular transformation was optimized. In an optimized reaction at elevated substrate concentration (100 mM; see the Extended Data Table 1 as an example), the presence of 1 – 5 mol% of NTP substrate for the kinases appears to be hardly of concern for the product isolation. Moreover, the substrates in Figure 3 are structurally quite different from ATP, so it would seem that they aren't really so difficult to separate from ATP/ADP. HPLC purification may be used in certain cases but just a

normal chromatographic separation will probably be fine in synthesis. And purification will be required in cases B and C (see line 201 and following).

An advantage of the kinase reaction (Figure 1B) is ignored. The activities of the individual enzymes can be adjusted to avoid accumulation of intermediates. Figure 4 shows the formation of product mixtures. The composition of the mixture varies with the substrate used. This renders the synthetic method unattractive in the opinion of this reviewer.

The cascade phosphorylation uses three different donor substrates for phosphate: pyrophosphate, polyphosphate and acetyl phosphate. A complex mixture of side products is generated which cause significant burden on the product isolation. Even the control of the reaction is complicated. Pyrophosphate and polyphosphate form complexes with metal ions required for enzyme activity. Polyphosphate is usually not well defined structurally as obtained commercially and the kinases differ in their ability to use the polyphosphate to reasonable completeness. The reviewer raises these points to emphasize that scheme C is not in fact a simpler reaction in practice than scheme B.

It has been known that kinases show promiscuity in the phosphorylation of NMP and NDP. For example, pyruvate kinase can work with different NMP and NDP substrates. Here, the authors show promiscuous reactivities of polyphosphate kinases and acetate kinase. It appears that they add acetate kinase because the polyphosphate kinase is insufficiently active to convert NDP into NTP. The characterization of polyphosphate kinases (Extended Data Table 2) is not sufficient. They show conversions after 18 h of incubation at large enzyme loading. Presumably the polyphosphate kinases are by less active than kinases typically used in these phosphorylation reactions.

Engineering of the phosphatase PhoC has yielded enzymes with higher activity for the nucleoside substrates. However, while interesting the development is incremental to earlier works of Asano and others. The role of the protein structures in the overall context of the study was not clear. It was not clear how the mutations have affected the enzyme activity. Moreover, the authors appear to gloss over critical issues of the reaction of the phosphatase in diversity-oriented synthesis. The outcome of the reaction with different substrates is extremely difficult to predict/control. Extended Data Table 1 illustrates this point. In reactions of NTP kinases, adding more enzyme is usually sufficient to overcome limitations of enzyme reactivity. In the transphosphorylation, the complex interplay of transfer and hydrolysis reactions requires optimization in a substrate-specific manner.

Overall, the reviewer felt that the study was better suited for a different journal within NPG (e.g., Communications Chemistry) or elsewhere (e.g., Green Chemistry).

Version 1:

Reviewer comments:

Reviewer #1

(Remarks to the Author)

The authors have addressed all of my concerns, so I can now recommend acceptance of this manuscript. Very nice work!

Reviewer #2

(Remarks to the Author)

The revisions the authors have made have satisfied my previous comments and, along with the additional revisions requested by the other reviewers, have significantly strengthened the manuscript. I support publication.

Reviewer #3

(Remarks to the Author)

Authors have responded to the comments of my original review. They disagree on several points raised. The reviewer retains the criticism that the overall cascade reaction is not convincing for production. Important elements of the proposed cascade reaction have been demonstrated in earlier studies.

Authors justify their efforts in engineering PhoC by the lack of broadly specific nucleoside kinases. In particular, they mention nucleosides modified at the 2'- and 3'-O position. The authors will know the field better than the reviewer, but only a very quick literature search revealed paper (Catalysts 2022, 12, 1401) describing broadly specific nucleoside kinases tolerating -O-Me substitution at both positions. The engineered PhoC may be a valuable addition to the toolbox of enzymes for phosphorylation, but as already mentioned in the first review it was not clear that this enzyme was a breakthrough. Information gained from structural analysis of the engineered phosphatase was rather low. Based on simple docking analysis in the experimental protein structure a flexible loop was suggested to play a role in substrate recognition and reaction selectivity.

Authors claim that NTP synthesis free of ATP represents a major advance. The point was not clear. The NTP solution from enzymatic production (≥ 50 mmol/L) will contain ATP in low amount, probably not more than 5 mol%. It was not evident which applications in RNA synthesis would be compromised by this level of contamination. However, if a specific

application of NTP did not tolerate any ATP, it would always be possible to run the kinase reaction on the nucleoside in the presence of catalytic amounts of the synthetic NTP. This would avoid ATP altogether. The small amount of NTP required to start the reaction might be synthesized chemically. The phosphatase cascade reaction is therefore not the only option to exclude ATP.

Figure 1A does not describe the state of the art correctly. Various papers show phosphorylation of NMP and NDP substrates by PPK. The PPK reactions do not require extra ATP as it is shown in the figure. For example, paper by Li et al. (ACS Synth. Biol. 2023, 12, 1772–1781) discusses the use of different PPKs for NMP conversion into NTP in the absence of ATP. Various PPKs were evaluated in this study. The conversion of NDP into NTP by other kinases often used for nucleotide donor regeneration (pyruvate kinase, AcK) is known.

Authors claim that the enzymatic cascade reaction is efficient for synthesis and might even be useful for production as the title implies. The reviewer disagrees. The syntheses of 2'-MOE-ATP and 2'-F-ATP involve complicated multistep procedures. Mixture of the phosphatase reaction must be filtered to remove enzyme and the pH be readjusted by titration and addition of a different buffer. Pyrophosphate must be precipitated by adding 750 mmol/L (!) of MgCl₂ and mixture be centrifuged. The reaction is then performed in the presence of PPK2 using hexametaphosphate (20 mg/mL; about 33 mmol/L assuming mass of 611 Da). Dilution up to this point was around 4-fold. After the PPK reaction the mixture is again filtered and further diluted. Reaction of AcK involves 50 mmol/L acetyl phosphate. The composition of the product solution is not stated but it will contain a lot of monomeric and oligomeric phosphate, Mg²⁺ and acetate/acetyl phosphate. Additionally, the product is a mixture of NTP (sometimes below 50% of total product; see Figure 4), NMP, NDP and nucleoside. The reviewer expressed concern about the purification required. Authors respond that a single "standard ion exchange chromatography" was sufficient to isolate the product. The purification was only superficially described. The point was not about whether the compounds can be fractionated on an ion exchange resin in principle. This is well documented in the literature. The point was about how practical is this purification (e.g., dilution of the target NTP, contamination of NTP by salt needed for elution and later removal of the salt etc.) considering the authors' claim that a new method of NTP synthesis, applicable for production, is presented.

Considering mixture of products formed (NTP, NDP, NMP, nucleoside) the reviewer asked about control of the product formation by adjusting the enzyme activities. A true one-pot transformation based on coupled reactions would allow for such control. Authors respond that their reactions could also be optimized by adjusting the individual enzyme activities. But this is not true. Their enzymatic reactions are performed uncoupled one from another. Only one enzyme is present at a time. Advantage of reaction coupling in cascade transformations is lost. The reaction of the phosphatase is not compatible with the reactions of PPK and AcK for several reasons, foremost the different pH requirements.

The kinase reactions are supposed to operate under thermodynamic control. Variation in composition of the product mixture was therefore not clear. Is this dependent on the ability of the enzymes to use certain NMP and NDP substrates?

The additional experiments showing use of the NTPs in oligonucleotide synthesis are appreciated. However, the evidence is demonstration of robustness of their method of DNA synthesis rather than demonstration of the efficiency of their NTP production.

Other points

Phenyl phosphate appears to be unattractive as donor for phosphorylation in production.

Response to Reviewers comments

We would like to thank the reviewers for their constructive feedback on our previous submission. We have now uploaded a revised version of the paper to the Nature Communications website that addresses the issues raised by the reviewers and editors. The specific changes made are outlined in the reviewer response below and are also highlighted in red in the marked-up versions of the manuscript and supporting information.

Reviewer #1:

Reviewer: This manuscript describes an ATP-free biocatalytic approach that can be utilized to convert nucleosides of diverse chemistry into their corresponding nucleoside triphosphates (NTPs). To start the pipeline, the authors engineered a non-specific acid phosphatase (PhoC) to synthesize nucleoside monophosphates (NMPs) using inorganic pyrophosphatase as the phosphate donor. The most promising variant (PhoC_4), obtained by combing beneficial mutations identified by screening variants in microtiter plates, achieves 96% conversion of 2'-MOE-A into 2'-MOE-AMP. Structure determination studies and molecular modeling of PhoC_4 suggest that the flexible loop plays an important role in substrate recognition. Importantly, PhoC_4 exhibits improved activity across a range of nucleosides relative to the starting variant (PhoC_1). An optimized enzymatic cascade was then established utilizing PPK and Ack to convert NMPs into NTPs, with the successful demonstration of nine structurally diverse examples from their starting nucleoside. The enzymatic pathway was scaled, under optimized conditions to generate >150 mg of 2'-MOE-ATP, which was used to produce a poly-2'-MOE-oligonucleotide pentamer. Overall, the manuscript is interesting, but will require significant revision prior to publication. The main concern is that certain areas of the manuscript are underdeveloped, lacking appropriate context, discussion, and referencing. Other areas contain jargony terms that should be avoided.

Response: We thank the reviewer for their positive and constructive feedback, and for their recognition of the significance of our work.

Reviewer: Important details in the results and discussion section are lacking. Some examples include: (i) undefined statements such as “high conversion” on line 75; (ii) theoretical size of the library; (iii) number of clones evaluated; and (iv) a clear description of the model in Figure 2E as a chimera from prior structures.

Response: We have now revised the manuscript text to address the reviewer comments. Specifically, we have removed the term ‘high conversions’ on line 75 and added details of the theoretical library size in the main text and Supplementary Fig. 1. The number of clones evaluated is already indicated in the text ‘following evaluation of more than 3,500 clones’ and Supplementary Fig. 1. The PhoC_4 model shown in Figure 2E was generated using molecular dynamics simulations. Here the loop (residues 151-159) closed to adopt a conformation similar to that observed in 1EOI. Additional details have been added to the main text.

Reviewer: Nonstandard terms should be removed from the text. This includes “holistic biocatalytic approach”, “telescoped fashion”, “vide infra”, and “pleasingly”.

Response: We have decided to keep one instance of the term ‘telescoped’ as it is a commonly used term, however we have added additional text to clarify its meaning. “*Ideally these enzymes would also be able to operate in a telescoped process, where consecutive biotransformations are performed without isolation or purification of intermediates.*” All other terms have been removed in line with the reviewer’s suggestion.

Reviewer: The manuscript lacks adequate citations. Some examples include: (i) the MOE chemical modification (line 95); (ii) the synthesis of 100 g of 5'- IMP (line 86); and the entire methods paragraph on the crystal structure which fails to cite any of the programs used for structure refinement.

Response: We apologize for this oversight and are grateful to the reviewer for highlighting this. Additional references have been included in the revised manuscript.

Reviewer: The source of compound 1a is missing. Was this synthesized or purchased? If it was synthesized, the details are needed.

Response: Compound 1a was purchased from Biosynth, this information was provided within the 'materials and methods' section in the original manuscript.

Reviewer: The synthesis of a pentamer of 2'-MOE-modified polyA is not a suitable demonstration of an oligonucleotide therapeutic. The language around this example should be tempered. Discussion about the length versus T_m values of the extended product is warranted.

Response: In line with the reviewer's suggestions, we have removed the claim 'showcasing a complete holistic biocatalytic approach to therapeutic RNA synthesis' from the abstract. We have reviewed our discussion on the synthesis of the 2'-MOE-modified polyA sequence and confirm that we have not described this sequence as an oligonucleotide therapeutic.

Regarding the discussion on the length vs T_m values, we are not clear on what the reviewer is requesting. We originally produced a 2'-MOE-modified pentamer as this is the typical length of the 2'-MOE modified flanking region of gapmers. However, in response to a comment from reviewer 3 we have now also produced a longer 8-mer sequence (in this case 2'-F-modified polyA) which has a higher T_m value (see response to reviewer 3 for further details). We note that in our previous work, we have produced sequences up to 21 nucleotides in length using our biocatalytic cascade (*Science* **380**, 1150–1154, 2023).

Reviewer: 7. The x-ray crystal structures of PhoC_1 and PhoC_4 warrant a more detailed structural comparison.

Response: We have now included additional discussion as requested.

Reviewer: The functional role of H168 and R183 should be empirically determined.

Response: We apologize there was a numbering error in our initial submission, these residues should be H168 and R201. We have now shown that mutation of R201 to alanine abolishes activity in PhoC_4, while a H168A substitution resulted in a 48-fold reduction in activity – this data is included in the revised draft.

Reviewer: The structural consequences of adaptive PhoC mutations should be discussed and shown in a figure, in particular E90, which is not located in the flexible loop.

Response: A comparative analysis of PhoC_4 with 1D2T and 1EOI suggests that the three mutations installed during PhoC_4 evolution (A90E, N151A and D154L) all reside in dynamic regions of the structure. N151A and D154L, reside on a poorly resolved loop region (residues 151-159) similar to that observed in 1D2T (residues 133-143). In 1EOI, which contains molybdate as a transition state analogue, this loop undergoes a structural rearrangement into a helical architecture to create a more occluded binding pocket. Based upon sequence and structural alignment, A90E in PhoC_4 is equivalent to residue S72 in 1EOI, which undergoes a 6 Å reorientation upon molybdate binding. We have now included a supplementary figure and additional discussion in the main manuscript.

Reviewer: Molybdate is mentioned in the text, but not in the structure.

Response: Molybdate was present as a transition state analogue in the previously reported structure of *E. blattae* acid phosphatase (pdb 1EOI) but is not present in our PhoC structures reported in this manuscript.

Reviewer: In certain places, figure callouts are unclear. For example, the substrates from Figures 2 and 3 are being called out as 1a, 1b, etc. It is not clear which figure they are referring to.

Response: We thank the reviewer for the comment – we have reviewed the manuscript text and ensured the figure callouts are clear and appropriate.

Reviewer: The comparison between the enzymatically produced poly-2'-MOE-A 5-mer and the authentic standard in Figure 5c should be more clearly described in the text.

Response: We have now included additional text as requested.

Reviewer: The manuscript would benefit from a summary of the strengths and weaknesses of the current strategy relative to other NTP synthesis approaches. Yields, time, purification, solvent, etc. would be useful information to share.

Response: We thank the reviewer for the suggestion. We have now expanded the conclusions to better emphasize some of the advantages of our approach, as well as highlighting some current limitations and opportunities for future improvements. Notably, since the original submission we have now shown that crude NTPs from our process can be used directly in enzymatic oligonucleotide synthesis, thus avoiding the need for NTP purification (Supplementary Fig. 7). It is unlikely that such an approach could be adopted with ATP-dependent methods, as even low concentrations of ATP contaminants severely compromise the purity of downstream oligonucleotide products (Supplementary Fig. 8).

Reviewer #2:

Reviewer: The manuscript by Meng et al. reports the development of a fully enzymatic, ATP-free cascade for the production of nucleoside triphosphates (NTPs), including chemically modified variants used in therapeutic oligonucleotides. This work elegantly builds on the seminal work in Science (DOI: 10.1126/science.add589), also by the Lovelock group, to address a key bottleneck in the RNA therapeutics field, namely, the scalable and selective synthesis of modified NTPs, which are often challenging to prepare chemically. The work describes a three-enzyme system comprising an engineered acid phosphatase (PhoC), polyphosphate kinase (PPK), and acetate kinase (AcK), which together enable phosphorylation of a wide array of canonical and modified nucleosides using inexpensive phosphate donors and without the need for ATP. This approach significantly simplifies purification by eliminating ATP and ADP impurities that would otherwise complicate downstream processes.

The authors demonstrate a well-executed protein engineering campaign on PhoC to enhance its activity on therapeutically relevant substrates such as 2'-MOE-adenosine. Directed evolution led to a triple mutant (PhoC_4) with an 8-fold improvement in conversion and an expanded substrate scope to several therapeutically relevant compounds. Structural characterization of PhoC_4 and docking of 2'-MOE-AMP provide insight into the mutational effects and highlight a loop region as likely contributing to substrate engagement and activity partitioning. This sets the stage for future engineering efforts and adds mechanistic depth to the study. Following this, a panel of PPKs and a promiscuous AcK were used to advance monophosphates to the triphosphate level across a range of substrates, achieving conversions up to 82%. The enzymatic synthesis of 2'-MOE-ATP on preparative scale and its subsequent use in oligonucleotide assembly via a polymerase-

endonuclease cascade further demonstrates the practical value of the method. The manuscript is clearly written, methodically executed, and of broad relevance. Overall, this is an excellent contribution to the field and represents a clear advance in enabling the biocatalytic synthesis of NTP building blocks. The work will be of interest to researchers in biocatalysis, nucleic acid chemistry, and RNA therapeutic development. As such I recommend publication after minor revisions below:

Response: We thank the reviewer for their positive comments.

Reviewer: While the engineering of PhoC_4 is thoroughly presented in terms of conversion data and structural interpretation, a kinetic comparison between PhoC_1 and PhoC_4 (e.g., k_{cat} , K_M values) would provide more quantitative context to the improvements and help evaluate trade-offs between rate and specificity.

Response: We are grateful for the reviewer's suggestion and have now carried out the kinetic comparison of PhoC_1 and PhoC_4. This data shows that directed evolution has led to a 10-fold increase in catalytic efficiency, due to a 3-fold improvement in k_{cat} and a 3.5-fold reduction in K_M . This information is now included in the revised draft.

Reviewer: The preparative scale synthesis is a great addition to the manuscript. It may also be helpful to include a brief comment on how the system behaves at larger scales - any limitations observed with respect to enzyme stability, precipitation issues, solubility, for example.

Response: We have now included additional text in line with the reviewer's suggestion.

Reviewer #3

Reviewer: Meng et al report on the phosphorylation of nucleosides by enzymatic cascade reactions. The resulting nucleoside triphosphates are substrates of engineered RNA polymerases in the synthesis of oligonucleotides. The authors use transphosphorylation catalyzed by an engineered phosphatase to produce the nucleoside monophosphates (NMP). The NMPs are phosphorylated sequentially to yield the nucleoside triphosphates (NTP). The nucleoside diphosphates (NDP) are formed as intermediates. The phosphorylations shown can be done chemically or through enzymatic cascade reactions, typically using NTP dependent kinases. The authors use enzymatic reactions independent of NTP and claim that NTP-free cascade transformations represent a major advance. This reviewer did not agree. The study is sound but the results do not constitute a significant improvement to existing methods of synthesis. The enzymatic reactions are all well-known and the developments are rather incremental.

Response: We respectfully disagree on several points raised by the reviewer. Scalable production of NTPs containing clinically important modifications remains a major challenge, and in fact the work contained within this manuscript is supported by diverse industry partners including leading suppliers of NTP building blocks and pharmaceutical end-users.

1. Relating to the advantages of ATP-independent synthesis. Although powerful in some instances, ATP-dependent methods have several drawbacks, and chemical synthesis remains the dominant method of modified NTP-production. Even when using ancillary enzymes for ATP recycling (which leads to additional costs), relatively high loadings of ATP are typically required. Furthermore, the typical stoichiometric phosphate donors used are far more expensive than pyrophosphate used for our PhoC biotransformations. Perhaps more importantly, when producing NTPs that are close structural analogues of ATP (e.g. 2'-F-ATP), the presence of ATP

and ADP by-products complicates downstream purification. In response to the reviewer's comments, in our revised draft we have now shown that crude NTPs produced using our biocatalytic process can be used directly in enzymatic oligonucleotide synthesis, avoiding the need for NTP isolation/purification (Supplementary Fig. 7). It is unlikely that such an approach could be adopted with ATP-dependent methods, as even low concentrations of ATP contaminants severely compromise the purity of downstream oligonucleotide products (Supplementary Fig. 8). Finally, we were unable to find any previous reports showing that ATP-dependent kinases are active towards 2'-OMe, 2'-MOE, LNA or 3'-protecting groups required for enzymatic synthesis of oligonucleotide therapeutics. In the revised manuscript, we have included an updated figure 1 and included additional text to better emphasize this important point. We note that we have evaluated a panel of ATP-dependent nucleoside kinases towards 2'-ribose modified substrates and detected no activity towards 2'-MOE or LNA nucleosides (data not shown).

2. Relating to the incremental nature of the study. Again, we respectfully disagree with the reviewer's commentary. While acid phosphatases have previously been shown to accept inosine and adenosine, to date there are no reports showing activity towards ribose modified substrates. Indeed, initial PhoC_1 activity towards the target modified substrates was very low making it unsuitable for direct use in NTP synthesis. Key to the success of our platform was the directed evolution of PhoC_4 which displays substantially improved catalytic efficiency towards 2'-MOE-A, the target substrate used for evolutionary optimization. Importantly, PhoC_4 also displays improved efficiency towards a diverse panel of clinically relevant substrates. The high degree of substrate promiscuity displayed by PhoC_4 is extremely valuable, and provides a common starting basis for engineering more efficient and specialized enzymes moving forward. We note that that nucleoside kinases do not appear to display this broad substrate tolerance towards ribose modifications.

We were also unable to find reports showing PPK or AcK activity towards diverse 2'- or 3'-modified substrates. Although PPKs have been used previously to produce nucleobase modified NTPs from the corresponding NMP, their applications have been limited by low conversions to the target NTP. This is the first report combining a PPK with an AcK to address equilibrium challenges and achieve high NTP conversions. Finally, we have now shown that the combination of an engineered PhoC, PPK and AcK can be used to produce important NTP building blocks needed to underpin biocatalytic production of RNA therapeutics.

Reviewer: Most importantly, complicated mixtures of nucleoside, NMP, NDP and NTP are produced (Figure 4) that require extensive purification before the NTP product can be used in further transformations. The argument used against the use of established NTP kinases, that the intermediate donor (e.g., ATP) must be removed as part of the product work-up, appears to be biased. The requirement for work-up of mixtures from reactions as shown in Figure 4 certainly exceeds the purification needed for a classical NTP-kinase cascade reaction. The effort in product purification required in each of these reactions depends on the specific transformation used and the extent to which the particular transformation was optimized. In an optimized reaction at elevated substrate concentration (100 mM; see the Extended Data Table 1 as an example), the presence of 1 - 5 mol% of NTP substrate for the kinases appears to be hardly of concern for the product isolation. Moreover, the substrates in Figure 3 are structurally quite different from ATP, so it would seem that they aren't really so difficult to separate from ATP/ADP. HPLC purification may be used in certain cases but just a normal chromatographic separation

will probably be fine in synthesis. And purification will be required in cases B and C (see line 201 and following).

Response: We strongly disagree with reviewer that the NTPs produced in this study require extensive purification. As shown in figure 5A we were able to isolate 139 mg 2'-MOE-ATP from our biotransformation using standard purification on DEAE Sepharose resin. Here, the NTP product was removed from minor NMP and NDP impurities, which are also generated in ATP-dependent kinase reactions. Furthermore, using ATP-dependent methods leads to additional ATP and ADP impurities which present additional purification challenges, especially when producing NTPs that are close structural analogues of ATP (e.g. 2'-F-ATP).

To further address the reviewer's concerns, we have now shown that enzymatically produced 2'-F-ATP in crude form can be used directly in biocatalytic oligonucleotide synthesis, thus obviating the need for NTP purification altogether (Supplementary Fig. 8). To our delight, crude 2'-F-ATP could be used directly in our polymerase and endonuclease cascade to furnish a poly-2'-F-A 8mer in 70% conversion. In contrast, efforts to produce poly-2'-F-A using 2'-F-ATP in the presence of low concentrations (from 1-10 mol%) of ATP contaminants gave complex product distributions arising from competing ATP and 2'-F-ATP incorporation (Supplementary Fig. 8).

Reviewer: Figure 1 is biased in the description of the different methods. While the reviewer agrees on some limitations of the chemical methods, these reactions aren't as low yielding and complex as the authors would like to suggest. The distinction made between reactions in B and C are not clear and appear to reflect the subjective preferences of the authors.

Response: We have now revised figure 1 in light of the reviewer's concerns.

Reviewer: An advantage of the kinase reaction (Figure 1B) is ignored. The activities of the individual enzymes can be adjusted to avoid accumulation of intermediates. Figure 4 shows the formation of product mixtures. The composition of the mixture varies with the substrate used. This renders the synthetic method unattractive in the opinion of this reviewer.

Response: While we appreciate the concentrations of individual enzymes can be adjusted in kinase reactions, the same is true of any enzyme cascade including ours developed within this study. Although we agree that there is some substrate-dependent variation in composition, this is to be expected – some of these substrates feature sterically demanding substituents and are challenging biocatalytic conversions. Moreover, the reactions have not been intensively optimized in each case to further drive higher conversions. Similarly, the enzymes have not been specifically engineered towards each substrate. Moving forward there are clear opportunities for efficiency gains using a combination of reaction and enzyme engineering. Finally, we note that a direct comparison to ATP-dependent kinases is simply not possible, as nucleoside kinases have not been shown to accept the ribose modified substrates used within this study.

To reflect the points above, we have now included additional text in the manuscript conclusions highlighting some current limitations of our approach and opportunities for future improvements.

Reviewer: The cascade phosphorylation uses three different donor substrates for phosphate: pyrophosphate, polyphosphate and acetyl phosphate. A complex mixture of side products is generated which cause significant burden on the product isolation. Even the control of the reaction is complicated. Pyrophosphate and polyphosphate form complexes with metal ions required for enzyme activity. Polyphosphate is usually not well defined structurally as obtained commercially and the kinases differ in their ability to use the polyphosphate to reasonable

completeness. The reviewer raises these points to emphasize that scheme C is not in fact a simpler reaction in practice than scheme B.

Response: The side products produced in the reaction (monophosphate, polyphosphate and acetic acid) did not present any challenges during product isolation (Figure 5A). Furthermore, the pyrophosphate and magnesium complex can be easily removed following the first step by filtration. Moving forward there are opportunities to engineer PhoC to operate efficiently using only 1 equivalent of pyrophosphate which would avoid the need for a filtration step. As described in the text, we have also shown that PhoC_4 and PPK can be used in a single enzymatic cascade if pyrophosphate is replaced by phenylphosphate, which does not coordinate to the metal ions.

Reviewer: It has been known that kinases show promiscuity in the phosphorylation of NMP and NDP. For example, pyruvate kinase can work with different NMP and NDP substrates. Here, the authors show promiscuous reactivities of polyphosphate kinases and acetate kinase. It appears that they add acetate kinase because the polyphosphate kinase is insufficiently active to convert NDP into NTP. The characterization of polyphosphate kinases (Supplementary Table 2) is not sufficient. They show conversions after 18 h of incubation at large enzyme loading. Presumably the polyphosphate kinases are by less active than kinases typically used in these phosphorylation reactions.

Response: As mentioned previously, we were unable to find any previous reports showing that ATP-dependent kinases are active towards 2'-OMe, 2'-MOE, LNA or 3'-protecting groups. Furthermore, we screened a panel of ATP-dependent nucleoside kinases towards 2'-ribose modified substrates and detected no activity towards 2'-MOE or LNA nucleosides.

As described in the text, polyphosphate kinases converted NMPs under thermodynamic control to mixtures of NDPs and NTPs. The acetate kinase (Ack) is used to phosphorylate any remaining NDP to improve yields of the target NTPs. The PPKs were initially evaluated using lyophilized cell-free extracts (Supplementary Table 2) so the exact enzyme loading is not known. However, for preparative scale reactions (2'-MOE-ATP and 2'-F-ATP) we used only 30 μ M purified PPK (0.06 mol%). This enzyme loading is comparable to that used in ATP-dependent kinase reactions (e.g. *Angew. Chem. Int. Ed.* 64, e202506330 (2025)).

Reviewer: Engineering of the phosphatase PhoC has yielded enzymes with higher activity for the nucleoside substrates. However, while interesting the development is incremental to earlier works of Asano and others.

Response: We strongly disagree with the reviewer. Previous studies involving PhoC have focussed on the production of inosine monophosphate (a flavour enhancer), AMP and phosphorylated aliphatic alcohols. As described previously in our response, we have been unable to find any reports showing activity towards nucleosides containing 2'-modifications and 3'-protecting groups required to produce RNA therapeutics. Moreover, we have shown that the starting PhoC variant had very poor activity towards our target substrates. Following our study, we now have access to an engineered PhoC variant that displays good activity towards a broad range of substrates featuring different bases, 1'- and 2'-modifications and 3'-protecting groups.

Reviewer: The role of the protein structures in the overall context of the study was not clear. It was not clear how the mutations have affected the enzyme activity.

Response: The loop region harbouring the N151A and D154L mutations is poorly resolved in the PhoC_1 and PhoC_4 structures, precluding any detailed structural comparison. However, molecular dynamics simulations suggest that the loop spanning residues 151-159 may reorder during substrate binding to adopt a more closed conformation, and this loop may play an important role in substrate recognition and/or the partitioning of phosphoryl transfer vs

hydrolysis activity. Thus, this structural characterization suggests targeting this loop region (151-159) for more extensive engineering should lead to the development of more proficient biocatalysts. In response to the reviewer's comment, we have added more detail to the structural characterization section, including information relating to the role of A90E.

Reviewer: Moreover, the authors appear to gloss over critical issues of the reaction of the phosphatase in diversity-oriented synthesis.

Response: We are not clear what the reviewer is suggesting. The focus of this study is the development of a low cost, sustainable method for NTP manufacturing. The connection to diversity-oriented synthesis is not immediately obvious.

Reviewer: The outcome of the reaction with different substrates is extremely difficult to predict/control. Extended Data Table 1 illustrates this point. In reactions of NTP kinases, adding more enzyme is usually sufficient to overcome limitations of enzyme reactivity. In the transphosphorylation, the complex interplay of transfer and hydrolysis reactions requires optimization in a substrate-specific manner. Overall, the reviewer felt that the study was better suited for a different journal within NPG (e.g., Communications Chemistry) or elsewhere (e.g., Green Chemistry).

Response: Predicting the outcome of any enzymatic transformation can be challenging. Extended Data Table 1 shows that high conversions can be achieved across a range of substrates at elevated substrate loadings. The different substrate and DMSO concentrations used in each reaction are a result of nucleoside solubility and are not related to the enzyme efficiency. We note that when exploring the substrate scope of enzymes, it is perfectly common to make adjustments to reaction conditions (e.g. time) to improve conversions. Finally, we note that nucleoside kinases have not been shown to operate on these substrates.

Response to Reviewer 3

Reviewer: Authors have responded to the comments of my original review. They disagree on several points raised. The reviewer retains the criticism that the overall cascade reaction is not convincing for production. Important elements of the proposed cascade reaction have been demonstrated in earlier studies. Authors justify their efforts in engineering PhoC by the lack of broadly specific nucleoside kinases. In particular, they mention nucleosides modified at the 2'- and 3'-O position. The authors will know the field better than the reviewer, but only a very quick literature search revealed paper (Catalysts 2022, 12, 1401) describing broadly specific nucleoside kinases tolerating -O-Me substitution at both positions. The engineered PhoC may be a valuable addition to the toolbox of enzymes for phosphorylation, but as already mentioned in the first review it was not clear that this enzyme was a breakthrough.

Response: The manuscript highlighted by the reviewer describes a nucleoside kinase with low level activity (24% conversion) towards 2'-OMe-uridine but it does not show conversion to the corresponding NTP product. As stated in our previous response we were unable to find examples of the use of ATP-dependent kinases in the synthesis of NTPs containing 2'-MOE and LNA modifications or 3'-protecting groups required for enzymatic synthesis of oligonucleotide therapeutics. In fact, in the paper highlighted by the reviewer, the authors evaluate 3'-acetyl protecting groups but detected no initial activity. This reference has been added to the revised manuscript.

Reviewer: Information gained from structural analysis of the engineered phosphatase was rather low. Based on simple docking analysis in the experimental protein structure a flexible loop was suggested to play a role in substrate recognition and reaction selectivity.

Response: We respectfully disagree with the reviewer. Our structural analysis highlighted key regions in the protein to be targeted in future engineering campaigns and will be of interest to researchers in the field.

Reviewer: Authors claim that NTP synthesis free of ATP represents a major advance. The point was not clear. The NTP solution from enzymatic production (≥ 50 mmol/L) will contain ATP in low amount, probably not more than 5 mol%. It was not evident which applications in RNA synthesis would be compromised by this level of contamination.

Response: The NTP solutions generated using our approach will not contain any ATP. To highlight the negative impacts of low-level ATP contamination on oligonucleotide synthesis, in our previous revision we performed polymerase catalyzed reactions using NTPs spiked with 1-10% ATP. Using NTPs containing only 1% ATP impurity led to a substantial (~3-fold) reduction in oligonucleotide yield and formation of impurities arising from misincorporation of ATP (Figure S8). This misincorporation of ATP will also likely be observed for enzymatic oligonucleotide synthesis using terminal deoxyribonucleotidyl transferases (TdT) and poly(U)polymerases (PUPs).

Reviewer: However, if a specific application of NTP did not tolerate any ATP, it would always be possible to run the kinase reaction on the nucleoside in the presence of catalytic amounts of the synthetic NTP. This would avoid ATP altogether. The small amount of NTP required to start the reaction might be synthesized chemically. The phosphatase cascade reaction is therefore not the only option to exclude ATP.

Response: We agree with the reviewer that this is another potential approach to NTP synthesis. Indeed, this has been shown by the Merck biocatalysis group who used 2'-F-(Sp)-thioATP (a close structural analogue of ATP) as their phosphate donor during synthesis of a cyclic dinucleotide. However, despite its close structural similarity to ATP, kinase activity towards the non-canonical donor was only modest and engineering was required to reprogramme cofactor selectivity. It remains to be seen whether ATP-dependent kinases can accept a wider range of modified NTPs as the phosphate donor. For example, as highlighted above we were unable to find examples of ATP-dependent kinases with activity towards 2'-MOE, LNA or 3'-protecting groups.

Reviewer: Figure 1A does not describe the state of the art correctly. Various papers show phosphorylation of NMP and NDP substrates by PPK. The PPK reactions do not require extra ATP as it is shown in the figure. For example, paper by Li et al. (ACS Synth. Biol. 2023, 12, 1772–1781) discusses the use of different PPKs for NMP conversion into NTP in the absence of ATP. Various PPKs were evaluated in this study. The conversion of NDP into NTP by other kinases often used for nucleotide donor regeneration (pyruvate kinase, AcK) is known.

Response: Figure 1A describes an ATP-dependent enzyme cascade and does not describe the use of PPKs. PPKs are described in Figure 1B: ATP-independent biocatalytic approach. We acknowledge that PPKs have previously been evaluated for activity towards nucleobase modified NTPs but were unable to find any examples PPKs with activity towards 2' and 3'-modifications. Additional text has been added for clarification.

Reviewer: Authors claim that the enzymatic cascade reaction is efficient for synthesis and might even be useful for production as the title implies. The reviewer disagrees. The syntheses of 2'-MOE-ATP and 2'-F-ATP involve complicated multistep procedures. Mixture of the phosphatase reaction must be filtered to remove enzyme and the pH be readjusted by titration and addition of a different buffer. Pyrophosphate must be precipitated by adding 750 mmol/L (!) of MgCl₂ and mixture be centrifuged. The reaction is then performed in the presence of PPK2 using hexametaphosphate (20 mg/mL; about 33 mmol/L assuming mass of 611 Da). Dilution up to this point was around 4-fold. After the PPK reaction the mixture is again filtered and further diluted. Reaction of AcK involves 50 mmol/L acetyl phosphate. The composition of the product solution is not stated but it will contain a lot of monomeric and oligomeric phosphate, Mg²⁺ and acetate/acetyl phosphate. Additionally, the product is a mixture of NTP (sometimes below 50% of total product; see Figure 4), NMP, NDP and nucleoside. The reviewer expressed concern about the purification required. Authors respond that a

single “standard ion exchange chromatography” was sufficient to isolate the product. The purification was only superficially described. The point was not about whether the compounds can be fractionated on an ion exchange resin in principle. This is well documented in the literature. The point was about how practical is this purification (e.g., dilution of the target NTP, contamination of NTP by salt needed for elution and later removal of the salt etc.) considering the authors’ claim that a new method of NTP synthesis, applicable for production, is presented.

Response: The process described in the manuscript has not been extensively optimized and we agree further process optimization would be required to enable scalable NTP production. These points are clearly discussed in the manuscript conclusions. For example, we suggest that optimization of individual enzymes towards target substrates of interest would minimize off-pathway processes (e.g. NMP hydrolysis by PhoC) and lead to enhanced productivity. To avoid the need for pH adjustments and dilutions there is the potential to develop enzymes with similar pH-optima. Alternatively, the NMP intermediate could be precipitated following the first step, avoiding the requirement for additional MgCl₂ and dilution. We have already begun process development for some selected NTP products and are working towards isolating the NMP intermediates and NTP products using precipitation and recrystallization. We have also shown that enzymatically produced NTPs in crude form can be used directly in biocatalytic oligonucleotide synthesis, obviating the need for purification altogether (Supplementary Fig. 8).

Reviewer: Considering mixture of products formed (NTP, NDP, NMP, nucleoside) the reviewer asked about control of the product formation by adjusting the enzyme activities. A true one-pot transformation based on coupled reactions would allow for such control. Authors respond that their reactions could also be optimized by adjusting the individual enzyme activities. But this is not true. Their enzymatic reactions are performed uncoupled one from another. Only one enzyme is present at a time. Advantage of reaction coupling in cascade transformations is lost. The reaction of the phosphatase is not compatible with the reactions of PPK and Ack for several reasons, foremost the different pH requirements.

Response: Although the reactions were performed sequentially in the preparative scale synthesis, we have also shown that PhoC₄ and PPK can be used in a single enzymatic cascade if pyrophosphate is replaced by phenyl phosphate as a donor (Supplementary Fig. 5). Both enzymes are active at pH 5.

Reviewer: The kinase reactions are supposed to operate under thermodynamic control. Variation in composition of the product mixture was therefore not clear. Is this dependent on the ability of the enzymes to use certain NMP and NDP substrates?

Response: For many substrates, the variation in the NDP to NTP ratio is small (Fig 4A). However, for some challenging substrates (e.g. 3'-aminoxy modified AMP) the

NTP yield was slightly lower suggesting the reactions have not yet reached equilibrium.

Reviewer: The additional experiments showing use of the NTPs in oligonucleotide synthesis are appreciated. However, the evidence is demonstration of robustness of their method of DNA synthesis rather than demonstration of the efficiency of their NTP production.

Response: The additional oligonucleotide synthesis reactions were designed to showcase the advantages of ATP-free synthesis. We highlight how even low concentrations of ATP contaminants severely compromise the purity of downstream oligonucleotide products (Supplementary Fig. 8). In contrast, crude NTPs produced using our biocatalytic process can be used directly in enzymatic oligonucleotide synthesis, avoiding the need for NTP isolation/purification (Supplementary Fig. 7).

Reviewer: Other points: Phenyl phosphate appears to be unattractive as donor for phosphorylation in production.

Response: It is not clear why the reviewer believes that phenyl phosphate is unattractive, but we respectfully disagree.